# Methods for the Synthesis of Phase Change Material Microcapsules with Enhanced Thermophysical Properties—A State-of-the-Art Review

**Refat Al-Shannaq \*, Mohammed M. Farid \* and Charles A. Ikutegbe**

Department of Chemical and Materials Engineering, The University of Auckland, Private Bag 92019, Auckland 1142, New Zealand; ciku619@aucklanduni.ac.nz
\* Correspondence: refat.alshannaq79@gmail.com (R.A.-S.); m.farid@auckland.ac.nz (M.M.F.)

**Abstract:** Thermal energy storage (TES) has been identified by many researchers as one of the cost-effective solutions for not only storing excess or/wasted energy, but also improving systems' reliability and thermal efficiency. Among TES, phase change materials (PCMs) are gaining more attention due to their ability to store a reasonably large quantity of heat within small temperature differences. Encapsulation is the cornerstone in expanding the applicability of the PCMs. Microencapsulation is a proven, viable method for containment and retention of PCMs in tiny shells. Currently, there are numerous methods available for synthesis of mPCMs, each of which has its own advantages and limitations. This review aims to discuss, up to date, the different manufacturing approaches to preparing PCM microcapsules (mPCMs). The review also highlights the different potential approaches used for the enhancement of their thermophysical properties, including heat transfer enhancement, supercooling suppression, and shell mechanical strength. This article will help researchers and end users to better understand the current microencapsulation technologies and provide critical guidance for selecting the proper synthesis method and materials based on the required final product specifications.

**Keywords:** microencapsulation; phase change materials; PCM; thermal energy storage

## 1. Introduction

In recent years, the energy storage sector has blossomed as an upbeat solution to increase the share of variable renewables. Its spectacular growth has centered primarily on electrochemical energy storage [1], given its application versatility in the power, buildings, and transport sectors. Generally, electrochemical energy storage faces cost limitations due to its shorter lifespans and difficulty in leveraging economies of scale in large volumes over a prolonged period. Other rising concerns include safety and supply chain geopolitical issues. Moreover, pumped storage hydropower [2], which relies on storage using water's potential energy, may avail better output as long as issues relating to costs resulting from terrain and limited available suitable locations are addressed. Additionally, while there is substantial evidence of hydrogen energy storage systems in the literature [3], large-scale projects remain in the development phase, and their high costs remain a concern. On the other hand, an alternative, reliable, secure, and flexible energy storage system, which has continued to gain interest and traction over decades, is thermal energy storage (TES) [4–6]. The array of in-front-of-the-meter-coupled TES knowhows introduce the possibility for demand shifting, seasonal thermal energy storage, sector integration, network management, and variable supply integration systems [7,8].

TES is a technology that leverage on the capability of a storage medium to trap copious amount of thermal energy either by heating or cooling for later use in various applications [9]. The thermal energy can be stored through any of the following: sensible heat, thermochemical, or as latent heat, in which the medium is phase change material

(PCM) [10]. TES systems play an important function in ameliorating the mismatch that exists between supply and demand, making them more attractive in terms of performance and reliability. Amongst the various energy storage techniques of interest, latent heat storage media based on PCMs has an advantage as it can store a significantly larger quantity of energy in less weight and volume of material in comparison with sensible heat storage systems. Moreover, it can absorb and release energy within a small temperature difference. Phase transition of PCMs accompanied either by absorption or releasing heat are commonly classified into solid–liquid, liquid–gas, solid–solid, and solid–gas [11]. PCMs exhibiting a solid–liquid phase transition experience a smaller change in volume during phase change transition compared to those undergoing liquid–gas transition. Similarly, they have the advantage of larger latent heat of storage than those PCM engage in solid–solid transition [12].

The solid–liquid PCMs can be either organic or inorganic compounds. During phase transition, they melt by absorbing heat and congeal by releasing stored heat preferably within a range as may be required by specific applications. For example, Esen and Ayhan (1996) [13] developed a mathematical model for estimating the stored thermal energy over time in a solar-assisted cylindrical latent heat PCM energy storage tank. They studied different types of PCMs with melting temperatures ranging from 60 to 70 °C. The model outcomes showed that the diameter of the storage tank is crucial for fast charging/discharging of PCMs. Additionally, the effective thermal conductivity, which reflects the rate of heat transfer, could be enhanced by inserting a thermal conducting material such as fins, lamellae, or matrix structures inside the storage tank. A few years later, Esen et al. (1998) [14] described two different geometric designs of PCM-embedded heating water storage tanks. The first mode, has the PCMs encapsulated in a cylindrical tube and packed into the tank and the heat transfer fluid (HTF) flow through the voids, while the second mode has PCMs filled into a tank and HTF flow through tubes packed into a storage tank. The results show that the heat transfer and system storage capacity vary according to cylindrical tank and tube diameters, volume of the PCM, HTF mass flow rate, and inlet temperature. Additionally, Esen M. (2000) [15] investigated, experimentally and theoretically, the thermal performance of a solar power heat pump system assisted with latent heat storage for space heating. The PCM was encapsulated inside cylindrical tubes and systematically placed into a storage tank, where the HTF flows between the tubes. The results reveal that for fast charging/discharging of PCM, as well as reasonable energy storage capacity, the thickness of the tube walls should be thinner in case of using multiple pipes with small radii. The supplied energy from the PCM storage tank is still insufficient for space heating. For better results, the heating space must be insulated well. Moreover, the use of the PCM slurry as a potential heat sink in electrical devices utilizing the latent heat of PCM during melting has recently gained more attention [16,17]. Roberts et al. (2017) [18] used different types of mPCMs, including metal coated and nonmetal coated mPCM slurry, in a microchannel heat exchanger. The result exhibits a major enhancement in HTF heat capacity when metal-coated mPCM slurry was used in compared to water at the same operating conditions. Additionally, the latent heat storage based on PCMs plays a significant role for improving system energy efficiency and shifting energy demand in other applications e.g., building [19], cooling of milk on farm [20], etc.

However, prior to usage, typical concerns such as leakage must be addressed through proper containment to prevent PCMs from exuding out from their host material. An effective and renown approach to conceal PCMs and prevent them from leaking is through microencapsulation [21]. Over decades, there have been many articles published on microencapsulation of PCMs using different methods, technologies, and types of PCM and shell. According to the literature, there is no comprehensive review on the most recent developments in PCM microencapsulation techniques, discussing their benefits and limitations and how these methods affect quality of the microcapsules. Therefore, this paper aims to cover and critically discuss the most recent developments, innovations, and adapted technologies used for PCM microencapsulation. This will include discussion on the use

of ultraviolet (UV) irradiation for rapid manufacturing of mPCMs. Different aspects of use for thermophysical enhancements of mPCMs, including heat transfer, supercooling suppression, and capsules mechanical strength, will also be discussed. This review paper will provide critical guidance for selecting the proper synthesis method and materials for PCM microencapsulation suitable for final product specifications, e.g., energy storage capacity, thermal conductivity, particle size and size distributions, capsule mechanical strength, etc.

## 1.1. Definition of Encapsulation

Encapsulation involves the engulfing of materials in a well-suited, thin, solid wall. The candidate material may be either solid, droplets of liquid, or gas. The material contained in the capsules represents the core, whereas that covering the core is often known as the shell wall [22] (Figure 1). Typically, encapsulated materials may fall under any of the following class based on their sizes, nanocapsules, microcapsules, and macrocapsules. For instance, microcapsules have a diameter within a range of 1 μm to 1 mm, while nanocapsules have diameters that are smaller than 1 μm. Macrocapsules possess diameters larger than 1 mm. The encapsulation of materials has evolved from examples in nature; the simplest example of a macrocapsule is a bird egg, and a cell along with its content is a microcapsule. The applications of microencapsulation are many. It commenced with the carbonless copy papers [23], whereby the top of the carbonless sheet is coated with dye or ink microcapsules and the bottom layer sheet is covered with a reactive clay. Over the years, the encapsulation technology has progressed and is now being applied in other fields such as the pharmaceutical [24], food [25], cosmetic [26], and textile industries [27], as well as the encapsulation of PCMs for TES applications [28].

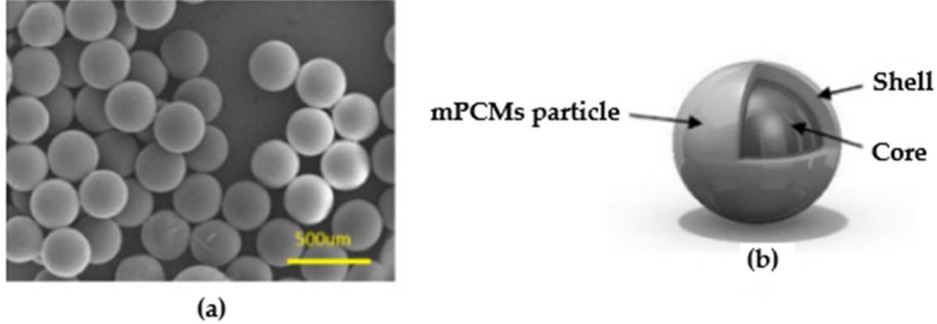

**Figure 1.** General view of (**a**) PCMs microcapsules and (**b**) core/shell microcapsules. Reused from Ran et al. (2020) [29].

## 1.2. Working Principle of PCM Encapsulation

Figure 2 illustrates the working principle of encapsulated PCMs. Initially, the PCM is in a solid state, and upon heating to temperatures above the PCM melting point, the encapsulated PCM absorbs the heat and starts melting while the shell material stays solid. The PCM temperature remains almost unchanged throughout the process of melting and the rate/amount of absorbed heat depends on the thermophysical properties of the selected shell and PCM materials. When the temperature drops below the PCM melting point, the PCM starts freezing, returns to its initial stage, and releases the absorbed heat.

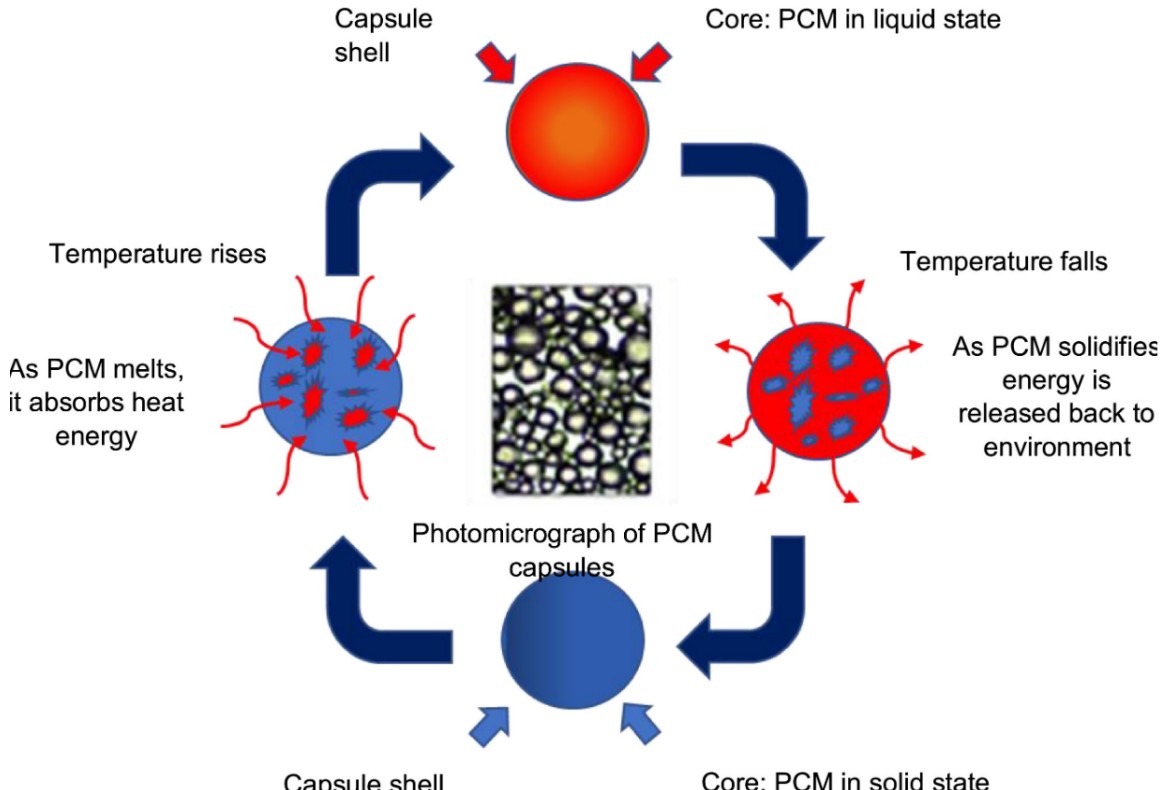

**Figure 2.** PCMs microscalpels' (mPCMs) absorbing/releasing energy cycle. Reused from Aslfattahi et al. (2020) [30].

*1.3. Morphology of PCM Microcapsules (mPCMs)*

The PCM microcapsules (mPCMs) may have a simple continuous core/shell, polypore capsules, continuous core capsules with more than one layer of shell material (microspheres morphologies), or matrix type capsules as shown in Figure 3. The morphology of the capsules depends on the core materials and the deposition process of the shell as well as the synthetic conditions. In the core/shell structure, the core material is in a continuous and concentered phase surrounded by protective continuous shell materials. The formation of this structure usually requires a two-step process; the formation of core materials (emulsification step) followed by a coating process (shell formation). However, the matrix structure of the microcapsules usually needs a one-step process, where the active material is dispersed in the structure and entrapped within the matrix material.

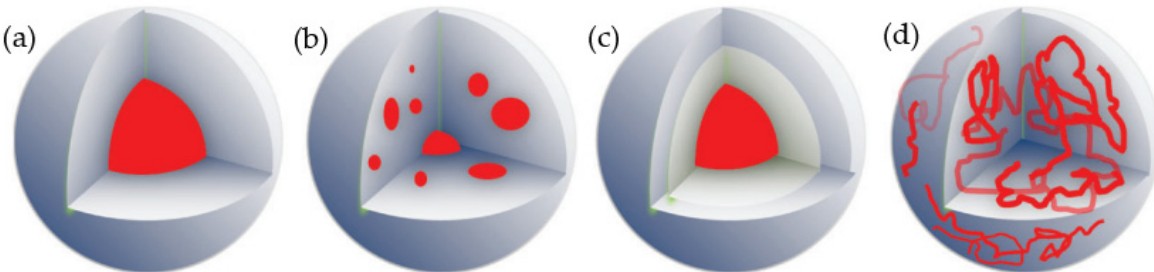

**Figure 3.** Schematic representation of cross section of four types of mPCMs morphologies: (**a**) continuous core/shell, (**b**) polypore capsules, (**c**) continuous core capsules with more than one layer of shell material, and (**d**) matrix type capsules. Reused from Trojanowska et al. (2019) [31].

The morphology structure has significant importance in terms of providing structural integrity and stability as well as its effect on the thermophysical properties of mPCMs. To enhance the mechanical strength of the mPCMs, continuous core capsule with more than one layer of shell material is preferred, but this will be at the expense of reducing heat storage capacity of core PCM. In contrast, the single-core-/-shell type provides high heat storage capacity but decreases the mechanical structural integrity and stability. In particular, the type of shell material and its mass proportion to core PCM defines the micro-PCMs' thermophysical and mechanical properties [32]. Lashgari et al. (2017) [33] observed that the type of shell material used contributes to the variation of internal morphologies of mPCMs, as shown in Figure 4. Consequently, the internal morphology of the mPCMs has an influence on their thermal behavior, where the formation of matrixlike morphology not only reduces the encapsulation efficiency, but also sacrifices the thermal cycling performance. On the contrary, multinucleus morphology would result in a proper thermal behavior.

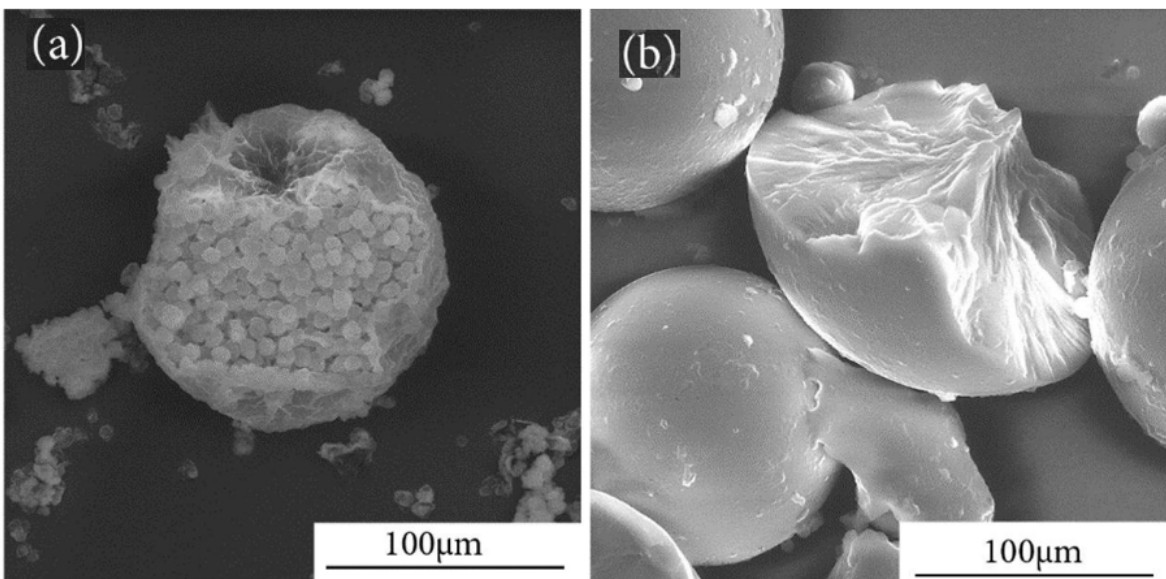

**Figure 4.** SEM micrographs of broken mPCMs: (**a**) multinucleus morphology for MMA-rich sample and (**b**) matrixlike morphology for BA-rich sample. Reused from Lashgari et al. (2017) [33].

## 2. Manufacturing Methods of mPCMs

Microencapsulation has already been proven as a successful technology in commercial applications, such as in the pharmaceutical and agrochemical industries [34], and recently in the textile industry [35] and in thermal energy storage applications [28]. Microencapsulation of PCMs is not only advantageous to mitigate the drawbacks of PCMs but also improves their thermophysical properties. For example, the heat transfer rate for PCM charging/discharging is significantly enhanced when PCM is microencapsulated [36]. In general, microencapsulation may be categorized into four groups, namely, physical–mechanical, physical–chemical, chemical–mechanical, and chemical processes. Table 1 summarizes the methods used for PCM microencapsulation with their subrelevant techniques and PCM/shell types.

**Table 1.** Summary of encapsulation methods, techniques, and PCM/shell types. n/a: not available.

| Category | Methods | Common Shell | PCM | Ref. |
|---|---|---|---|---|
| Chemical | In situ polymerization | MF, UF, PEG modified MF | Organic/ inorganic | [37–41] |
| | Interfacial polymerization | PU, TDI, DETA, Polyamide | Organic/ inorganic | [42] |
| | Suspension polymerization | PMMA, PBMA, PS, PDVB | Organic | [43–47] |
| | Emulsion/mini emulsion polymerization | PMMA, PS, SBA, PS-MMA | Organic/ inorganic | [48,49] |
| Physical– chemical | Coacervation | Gum Arabic/gelatine, agar agar/gum Arabic, UF, MF, UMF, Chitosan/gum Arabic | Organic | [50–52] |
| | Sol–gel method | $SiO_2$, $TiO_2$, $Fe_3O_4/SiO_2$, $PMMA/SiO_2$, $SrTiO_3$ | Organic/ inorganic | [53–57] |
| | Self-assembly | $CaCO_3$, $Cu_2O$, $TiO_2$ | Organic | [58–61] |
| | Ionic gelation | | Organic | |
| Physical– mechanical | Spray drying | Gelatine acacia, LDPE, EVA | Organic | [62,63] |
| | Solvent evaporation | PVC, ethyl cellulose, MMA-co-HEMA | Organic | [64–66] |
| | Vacuum impregnation | n/a | Organic | n/a |
| | Fluidized bed | Acrylic | Organic/ inorganic | [67] |
| | Pan coating | n/a | Organic | n/a |
| | Air-suspension coating | n/a | Organic | n/a |
| | Vibration nozzle | n/a | Organic | n/a |
| | Centrifugal extrusion | n/a | Organic | n/a |
| Chemical– mechanical | Microfluidic method | Silicone, PDMS | Organic | [68–70] |
| | Melt coaxial electrospray method | Sodium alginate | Organic | [71] |

## 2.1. Chemical Methods

Chemical methods utilize polymerization or a condensation process of monomers, oligomers, or prepolymers as raw materials to form shells at an oil–water interface. Commonly, chemical methods include (a) in situ polymerization, (b) interfacial polymerization, (c) suspension polymerization, and (d) emulsion/miniemulsion polymerization. Figure 5 represents the shell schematic formation of mPCMs using different chemical methods. The differences between these four polymerization methods mainly depend on the location where the polymerization takes place. In the in situ polymerization, the prepolymer, which is usually melamine-formaldehyde (MF) or urea-formaldehyde (UF), is cross-linked at oil–water droplet interfaces to form a continuous shell around the PCM droplets (Figure 5a); however, in the interfacial polymerization method, the shell forms due to the reaction between a water-soluble monomer and an organic-soluble monomer at the oil–water in-

terface (Figure 5b). In contrast, the polymer particles form within a PCM droplet in the presence of an organic-soluble initiator and then precipitate out at the oil–water interface in the case of suspension polymerization (Figure 5c). The emulsion polymerization method involves combining the monomer with a surfactant in an oil phase, while the initiator is dis-solved in the water phase. Because the initiator is only present in the aqueous phase, the polymerization reaction begins outside of the droplets and micelles and subsequently progresses to the micelles (Figure 5d). More details of chemical methods used for microencapsulation of PCMs with the most relevant published work will be discussed in the following subsections.

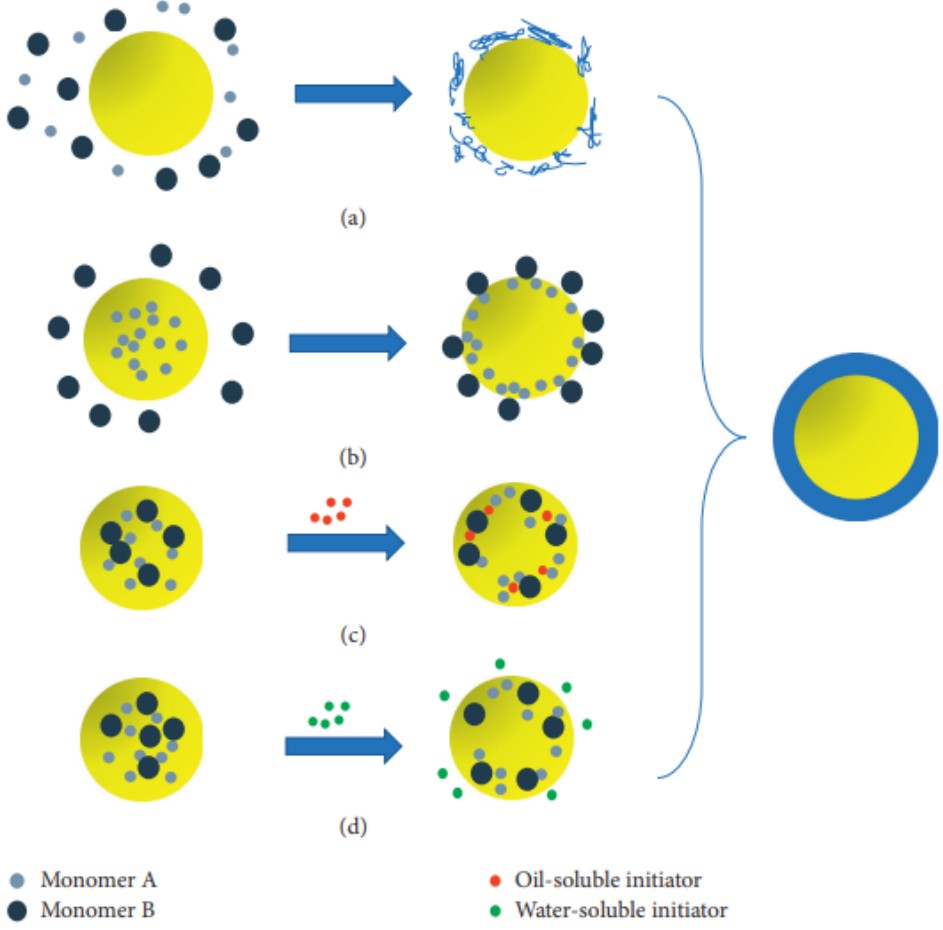

**Figure 5.** Schematic diagrams of mPCMs formation using: (**a**) in situ polymerization, (**b**) interfacial polymerization, (**c**) suspension polymerization, and (**d**) emulsion/miniemulsion polymerization. Reused from Peng et al. (2020) [72].

### 2.1.1. In Situ Polymerization

The most common example of this method is the condensation polymerization of urea or melamine with formaldehyde to form cross-linked UF or MF capsule shells. In this method, droplets are first formed by dispersing the core material (PCMs) into an aqueous phase containing a small fraction of emulsifier, followed by the addition of proper monomers or prepolymers of urea with formaldehyde or melamine with formaldehyde. After the pH of the system is lowered, the polycondensation reaction starts, yielding cross-linked urea–formaldehyde or melamine–formaldehyde resins. When the resin reaches a high molecular weight, it becomes insoluble in the aqueous phase, precipitates out, and deposits at the oil–water interface of the droplets. Subsequently, the resin hardens to form microcapsule shells, as depicted in Figure 6.

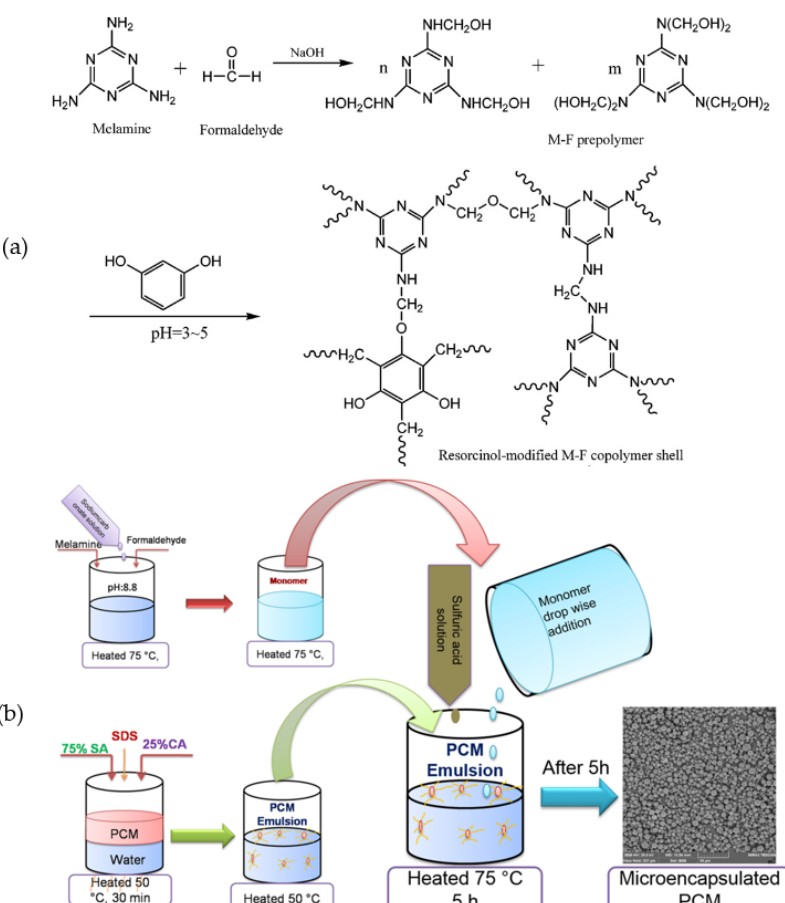

**Figure 6.** (**a**) MF prepolymer polymerization reaction and (**b**) schematic presentation of stepwise preparation of mPCMs with fatty acids eutectic mixture as core and MF as shell using in situ polymerization. Reused from Srinivasaraonaik et al. (2020) [37].

Several articles have been published over the last decade on the microencapsulation of PCMs using in situ polymerization [73–78]. Konuklu et al. (2014) [79] synthesized mPCMs containing caprylic acid PCM with different wall materials, including UF, MF, and urea–melamine–formaldehyde (UMF) resins. SEM tests revealed that the mPCMs having melamine in their outer walls dominated the spherical structure. However, lower latent heat storage capacity was obtained when PMF or PUMF was used for outer wall material instead of PUF. The caprylic acid/PUF mPCMs enthalpy is 93.9 J/g, corresponding to 59 wt.% of PCM mass content. Han et al. (2020) [38] prepared mPCMs with paraffin as the core material and MUF polymer as the shell material under different processing conditions. The optimal operating conditions were found when the core-to-shell ratio was 1.5, concentration of SMA was 20 mg/mL, and reaction temperature, reaction time, and final pH values were 80 °C, 2 h, and 5.5, respectively. Under these conditions, the mPCMs' melting enthalpy was found to be equal to 134.3 J/g, corresponding to an encapsulation efficiency of about 77.1%. Srinivasaraonaik et al. (2020) [37] synthesized mPCMs with fatty acids eutectic mixture (75% SA + 25% CA) as the core and MF as the shell using in situ polymerization. The obtained results reveal that at the optimized pH (3.2) and agitator speed (1500 rpm), the microcapsules possess smooth surface morphology and are spherical in shape with particle sizes of approximately 10 μm. Furthermore, the encapsulation efficiency and latent heat of fusion of mPCMs were 85.3% and 103.9 kJ/kg, respectively. Kumar et al. (2021) [39] prepared mPCMs through a facile in situ polymerization process using 1-dodecanol as the core PCM and the MF as the shell material. The results revealed that the capsules have surface roughness with no chemical interaction between the core and the shell materials. The TGA results showed that the mPCMs start losing weight at

132.8 °C, which was higher than that of the pure PCM. Furthermore, the mPCMs exhibited moderate storage capacity of 79.5 kJ/kg but with excellent thermal reliability (reliability index of 94.4%).

Furthermore, Zhang et al. (2022) [80] fabricated 4 µm capsules with MF resin shell via cellulose nanocrystal (CNC)-stabilized Pickering emulsion in situ polymerization. The phase change enthalpy of commercial paraffin wax and n-octadecane microcapsules are as high as 164.8 and 185.1 J/g, corresponding to PCM core material content of 87.0 and 84.3%, respectively. The results also showed that the mPCMs exhibit good thermal reliability during the phase change process, since they displayed almost the same phase change enthalpy after 200 heating/cooling thermal cycles, and the retention rate of ΔH reaches up to 99.7%. Moreover, the mPCMs are self-extinguishing due to the flame-retardant properties of the MF shell. Mustapha et al. (2022) [81] reported a range of emulsifiers and their role in the formulation of volatile core microcapsules via the one-step in situ polymerization process. The results showed that the emulsifiers functional group type plays an important role for final mPCMs' product quality, where emulsifiers having carboxyl groups speed up the reaction rate to the extent that the UF particles become too large to maintain controlled and steady deposition onto the O/W interface, leading to porous microcapsules. In contrast, the emulsifiers with hydroxyl groups created small UF particles leading to poor encapsulation. Intermediate reactions were displayed with amine/amide groups or a combination of groups. Notable emulsifiers with successful results include gelatine (GEL), methylcellulose (MC), xanthan gum (XG), chitosan (CHI), polyacrylamide (PAM), and poly(ethylenimine) (PEI). Figure 7 shows the effect of different cross-linking agents on the microcapsule morphology and core material retention. Incomplete microcapsules are formed when catechol and hydroquinone were used as crosslinkers in compared to resorcinol. Broken pieces of shell material were formed with absolutely no retention of the core material. In conclusion, the polymerization process was greatly hindered by introducing these alternate crosslinkers.

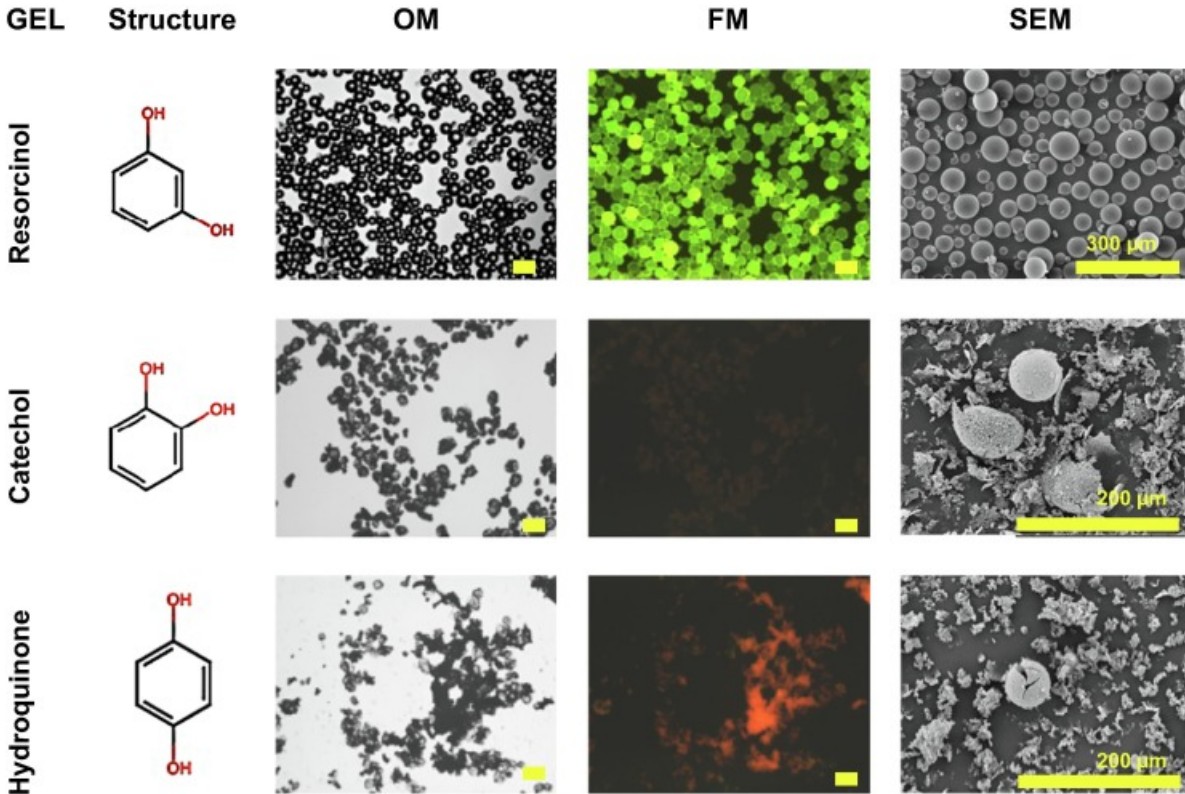

**Figure 7.** The OM, FM, and SEM images of the GEL mPCMs with different cross-linking agents, namely resorcinol, catechol, and hydroquinone. Reused from Mustapha et al. (2022) [81].

UF or MF resins are some of the commonly reported shell materials utilized for the fabrication of microcapsules through the in situ polymerization technique. However, due to their toxic nature, the production of formaldehyde-based microcapsules needs to adopt a significant preparation route coupled with stringent safety precaution measures. In addition, some of the residues of these formaldehyde resins in shells can cause environmental and health problems. Therefore, many researchers have been considering manufacturing mPCMs with low remnant formaldehyde content [82,83]. Sumiga et al. (2011) [84] fabricated MF mPCMs with ammonia as a scavenger for residual formaldehyde reduction. Furthermore, Su et al. (2011) [85–87] used a novel methanol-modified MF prepolymer as a shell material for fabrication of low remnant formaldehyde-content PCM microcapsules. Results show that this can reduce the free formaldehyde in shell material by increasing the cross-linking structure, enhance the resistance deformation of MF shell, and achieve a reasonable encapsulation efficiency of 85.4% with a PCM content of 71 wt.%.

### 2.1.2. Interfacial Polymerization

This method is characterized by wall formation via rapid polymerization of monomers at the surface of the droplets of dispersed core material. Droplets are first formed by emulsifying an organic phase consisting of core materials and oil-soluble reactive monomers, which are usually isocyanate or acid chloride, in an aqueous phase. By adding a water-soluble reactive monomer, rapid reaction happened between the two monomers at the interface of the droplets to form a polymer shell, as shown in Figure 8. Several articles have been published over the last few years on microencapsulation of PCMs using in interfacial polymerization technique [88–90]. Cai et al. (2020) [42] synthesized eco-friendly mPCMs via the interfacial polymerization of toluene-2, 4-diisocyanate (TDI), and diethylenetriamine (DETA). A facile solvent-free synthesis route was developed owing to the good compatibility of TDI and dodecanol dodecanoate, in which the cosolvent (cyclohexane) can be removed during the polymerization process. The as-prepared mPCMs were in the size scope from 10 to 40 μm and had a latent heat of fusion in the range of 103–140 J/g, which depends on the PCM to wall mass ratio. Nikpourian et al. (2020) [91] nanoencapsulated paraffin wax with polyurethane (PU) via the interfacial polymerization. Prior to encapsulation, the spherical and solid nanoemulsion of paraffin wax with control size distribution were prepared using the novel semisolvent–nonsolvent method in the presence of sodium dodecyl sulfate (SDS). Then, the obtained nanoparticles were encapsulated with a polyurethane shell, based on TDI. The field emission scanning electron microscopy (FESEM) confirms the formation of spherical paraffin wax nanoparticles with a particle size distribution of 25–185 nm. The melting enthalpy of prepared nanocapsules is 153.9 J/g, corresponding to 80.2 wt.% of core content. Furthermore, the nanocapsules exhibit good thermal reliability and stability after 100 heating/cooling cycles. Lone et al. (2013) [68] reported an easy and effective approach for fabricating highly monodisperse PCM polyurea microcapsules using a tubular microfluidic technique. Yoo et al. (2017) [92] successfully fabricated a microencapsulated PCM (methyl laurate) with poly(urethane) (PU) composite shells containing the hydrophobized cellulose nanocrystals (hCNCs) using an in situ emulsion interfacial polymerization process. Gao et al. (2021) [93] prepared chitosan-based polyurethane (c-PU) mPCMs using the interfacial polymerization reaction of hexamethylene diisocyanate and chitosan accompanied by the attraction-assisted charge. The mPCMs with a c-PU green shell exhibited reasonable heat storage capacity of 106.3 J/g with an equivalent PCM content of 71.4% and with excellent thermal stability and cyclic durability. The c-PU mPCMs with reversible photochromic properties show promising application in the fields of anticounterfeiting technology and flexible, wearable, UV-protective clothing.

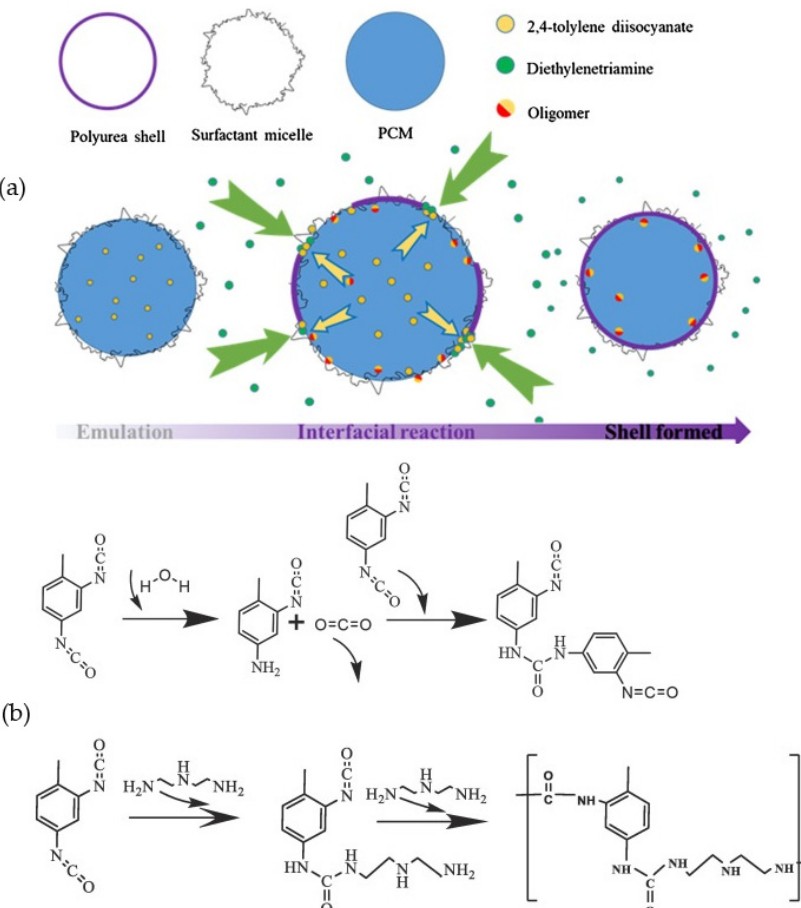

**Figure 8.** (**a**) Schematic diagram for the formation mechanism of mPCMs via the interfacial polymerization and (**b**) the reaction mechanisms of the interfacial polymerization. Reused from Cai et al. (2020) [42].

### 2.1.3. Suspension Polymerization

This method is a heterogenous radical polymerization process that is used for the production of many common commercial resins, such as polyvinyl chloride (PVC) [94], polystyrene [95], and poly(methyl methacrylate) (PMMA) [96]. Over time, suspension polymerization has been used comprehensively for the synthesis of functional microspheres [97] and for the fabrication of PCM microcapsules [43–46]. Figure 9a shows the formation process of mPCMs via free radical suspension polymerization. The oil phase, which consists of PCMs, water-insoluble monomers, and free radical oil-soluble initiators, is dispersed in the aqueous phase as droplets by high shear homogenization along with the use of small amounts of suspending agents. When the temperature reaches the decomposition temperature of the free radical initiator, the reaction starts to take place inside the droplets, and the generated polymer precipitates out of the PCM monomer mixture to form polymer particles. These particles continue to grow in number and size as polymerization continues and are deposited at the oil/water interface by the action of hydrophobicity to form the capsule shell, as shown in Figure 9b.

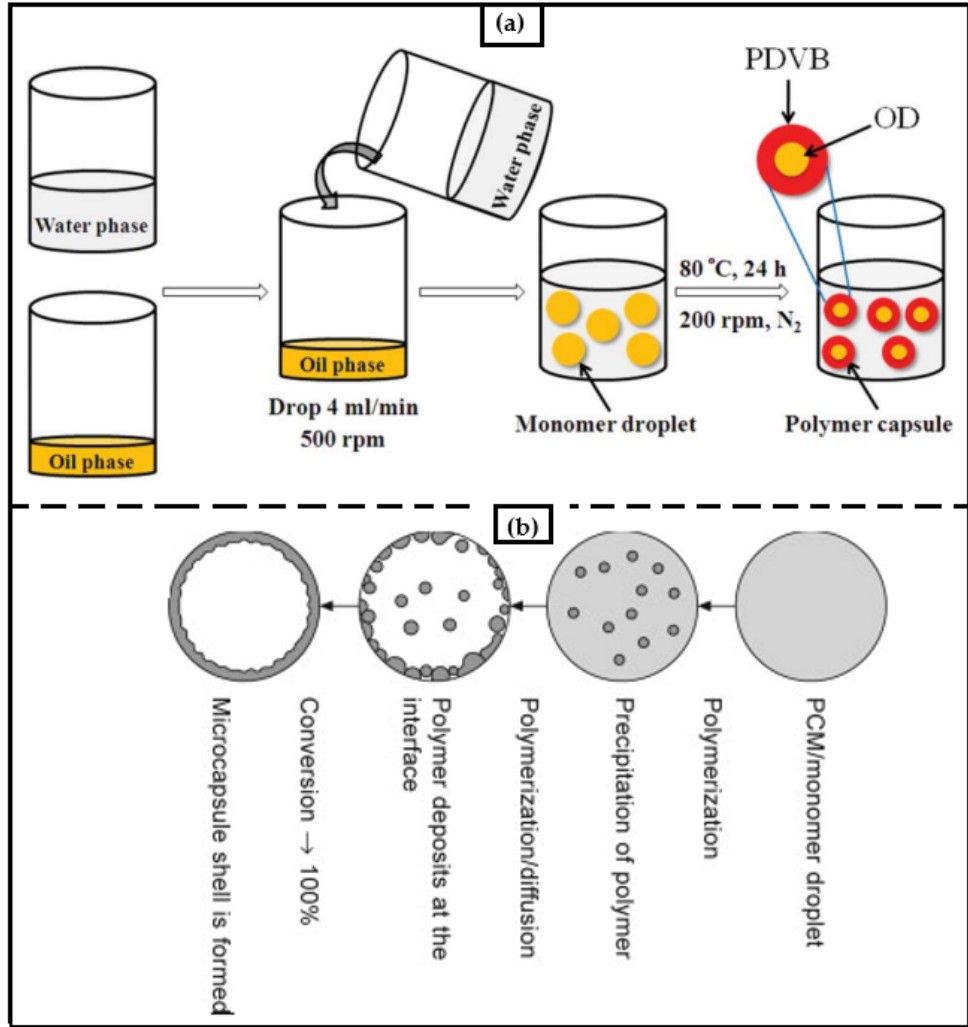

**Figure 9.** (**a**) Schematic of the preparation process of the PDVB/OD microcapsules by suspension polymerization (Reused from Chaiyasat et al. (2013) [47]) and (**b**) mechanism of capsule formation using suspension polymerization (reused from Al-Shannaq and Farid (2015) [28]).

In the suspension polymerization method, there are two sources of energy used to produce free radicals that initiate polymerization. (i) Thermal decomposition: the initiator is heated until a bond is homolytically cleaved, producing two radicals. This method is used most often with organic peroxides or azo compounds. (ii) Photolysis: in this case, several steps occur. They are: (a) a UV light of a certain spectra emission attacks and overlap the photo-initiator absorption spectra; (b) upon absorption, the photo-initiator molecule is promoted from the ground to either a single or triple excited electronic state; and (c) the excited molecule undertakes a cleavage (or reaction) with another molecule to produce initiating free radicals. In the following subsubsections, we will spot the light on the recent developments of mPCMs' synthesis using traditional thermal process and UV microencapsulation.

Thermal Microencapsulation

The choice of the ideal shell material for engulfing PCMs is central for manufacturing high performing mPCMs. Several polymers and copolymers have been extensively used to encapsulate PCM [43,44]. Chaiyasat et al. (2011) [98] and Supatimusro et al. (2012) [99] employed poly(divinylbenzene) as a shell material to encapsulate octadecane PCM using the suspension polymerization approach. Sánchez-Silva et al. (2011) [100] used polystyrene to encapsulate Rubitherm® RT31. Qiu et al. (2013) [101] prepared mPCMs us-

ing acrylic-based polymer shells. Qiu et al. (2013) [102] microencapsulated n-alkane with a p(n-butyl methacrylate-co-methacrylic acid) shell. Al-Shannaq et al. (2015 and 2016) [43,46] successfully synthesized cross-linked methyl methacrylate mPCMs via the suspension polymerization technique. Sánchez-Silva et al. (2010) [103] microencapsulated PCMs with shell materials made from styrene-methyl methacrylate copolymer. Similarly, with the suspension polymerization approach, microcapsules comprising of n-hexadecane as the core and PMMA and poly(butyl acrylate-co-methyl methacrylate) (poly(BA-co-MMA)) as the shells were prepared [33]. Furthermore, previous works have demonstrated that acrylic resins, such as PMMA, are promising shell materials to be used for microencapsulation of PCMs because they possess high mechanical strength, good chemical stability, are nontoxic, and are ease to use [104]. Al-Shannaq et al. (2015) [43] investigated the microencapsulation of paraffin (core) using PMMA (shell) by means of suspension polymerization. They found that by the utilization of mixed surfactants, long-term emulsion stability and microcapsules of regular spheres with smooth surfaces could be achieved. Sari et al. (2009) [105] encapsulated octacosane into a PMMA shell and reported that the microcapsules displayed good chemical stability and energy storage potential. They demonstrated that the Octacosane/PMMA capsules had an average latent heat of about 87.45 J/g during phase transition, with excellent thermal reliability even after 5000 thermal cycles. Thermal microencapsulation is an energy-intensive process since it is performed at a high temperature (70 to 90 °C) and needs a long rection time exceeding 4 h. Additionally, it is unfavorable for microencapsulation of low-melting-temperature PCMs. UV microencapsulation is an alternative to the thermal process since the reaction can occur at room temperature and within a shorter time.

UV Microencapsulation

Nevertheless, in comparison to traditional thermally induced processes, the use of UV photo initialization polymerization is becoming attractive due to its suitability for low-temperature microencapsulation of PCMs. Thermally induced polymerization is usually performed at higher temperatures, which could damage heat-sensitive PCMs. Beyond this, they require a considerable reaction time, especially when considered for large-scale production. In that sense, suspension polymerization with the UV irradiation-initiated approach offers a better alternative to thermal treatment by reducing the polymerization time and energy consumption [45]. The technique also helps to retain the thermophysical properties of temperature-sensitive PCMs. PMMA/paraffin microcapsules were fabricated by the continuous stirring of the oil/water emulsion placed in a low-columned quartz container for 30 min at a speed of 600 rpm and irradiated by a medium-pressure mercury UV lamp (1000 W). The results show that the latent heat of mPCMs is 165.01 kJ/kg, which is equivalent to 61.2 wt.% of core materials inside the microcapsules [106]. Similarly, a UV photoinitiated dispersion polymerization approach was used to microencapsulate a high melting paraffin wax (peak melting point of 56.3 °C) and MMA. The oil/water emulsion was irradiated by a medium-pressure mercury UV lamp having a power of 2500 W for 30 min. According to their findings, the optimal performing mPCMs retained up to 66 wt.% PCM in the shells, with an average latent heat of 109.6 J/g. Increasing the PCM content beyond this point led to poor encapsulation efficiency [107]. Wang et al. (2014) [108] studied the effect of UV light-initiated emulsion polymerization on the nanoencapsulation of stearic–eicosanoic acid with MMA. A medium-pressure mercury UV lamp of 2500 W rating was positioned at a height of 20 cm over the top of the low-column quartz container. The produced capsules have an average latent heat of about 127 J/g and melting temperature of 56.9 °C. According to the authors, both cationic and nonionic emulsifiers are suitable for obtaining high-quality nanocapsules. Zhang et al. (2017) [109] investigated the development of PMMA/stearic acid microcapsules through UV-initiated emulsion polymerization using iron (III) chloride as the photosensitive reagent. A medium-pressure mercury UV lamp having 2500 W was used. Other studies in this area have encapsulated PCMs of a higher melting temperature with an encapsulation efficiency not exceeding

70% [110]. Investigators from The University of Auckland (New Zealand) established a rapid, scalable, and energy-efficient way for the low-temperature microencapsulation of PCMs using both thin film UV reactor [111] and coiled-tube UV reactor [45]. Figure 10a,b show the schematic representation and experimental set up of the UV coiled-tube reactor. In this instance, the prepared emulsion was introduced into a holding tank and purged with $N_2$ gas at 60 bubbles/min during the photopolymerization reaction. A peristaltic pump was used to recirculate the emulsion at different volumetric flow rates from the holding tank to the coiled tube reactor. The influence of two mercury lamps (medium pressure) having power ratings of 450 W and 250 W on the process parameters were studied. Each lamp was placed at the center of the reaction cell to photo-induce the polymerization reaction of MMA. When using a suitable organic soluble photoinitiator such as Irgacure 918, the optimum encapsulation yield, conversion, encapsulation efficiency, and PCM content were achieved after only 10 min of polymerization. According to their findings, a higher emulsion flowrate (of 31.2 L/h) offers better PCM encapsulation in terms of both PCM content and encapsulation efficiency. Consequently, increasing the PCM-to-monomer mass ratio up to 2:1 led to a corresponding increase in the latent heat of microencapsulated PCMs, with ir-regular spherical shapes of microcapsules. In summary, the study showed that photo-induced polymerization with UV radiation represents a better alternative to thermal treatment by reducing polymerization time and energy consumption.

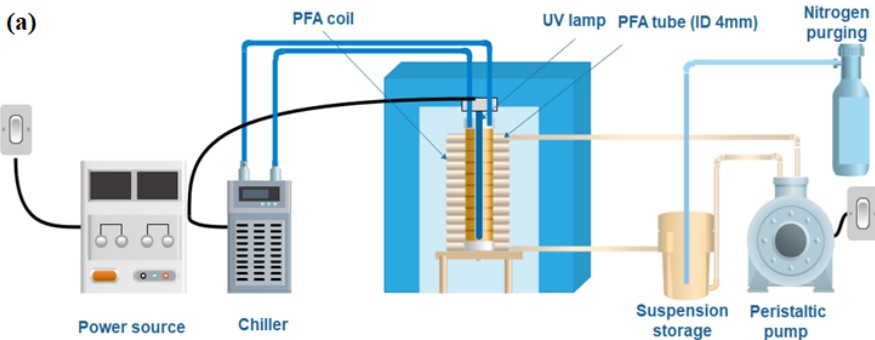

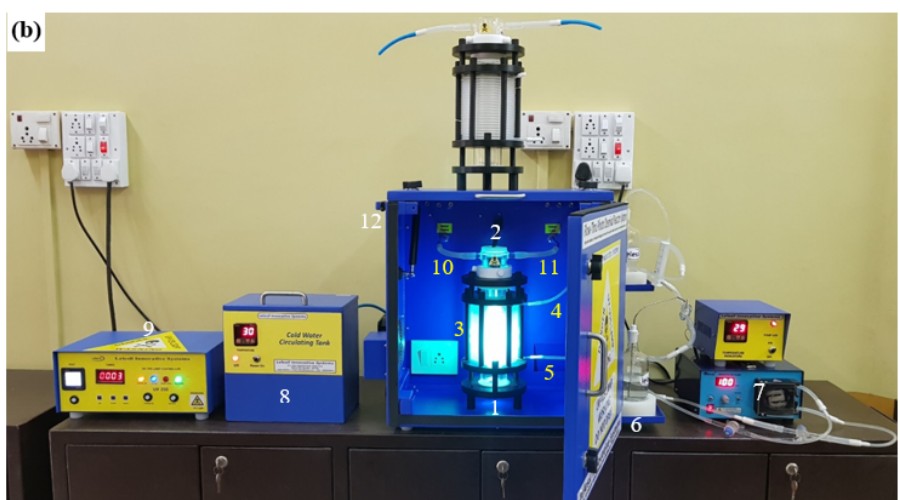

**Figure 10.** (**a**) Schematic diagram, and (**b**) experimental set-up of UV coiled tube reactor: (1) UV reaction cell, (2) UV lamp, (3) coiled PFA tube (ID:4mm), (4) inlet emulsion to the UV reaction cell, (5) outlet emulsion from the UV reaction cell, (6) circulation storage tank, (7) peristaltic pump, (8) chiller, (9) power supply, (10) inlet cold water, (11) outlet hot water and (12) UV protecting box. Reused from Ansari et al. (2021) [45].

In a recent study by Ikutegbe et al. (2022) [112], the microencapsulation of a much lower melting PCM, such as PureTemp® PT 6, was investigated using the UV coiled-tube reactor illustrated in Figure 10. The low temperature PCM microcapsules may be mixed with a suitable fluid and applied as an efficient heat transfer slurry in air handling units or used to form parts of walls and ceiling components of refrigeration trucks or cold chambers for the preservation of perishable food. Being able to satisfy the cost and quality of perishable food products through the deployment of these type of mPCMs gives economic value not just to the producers of perishable foods but also to retailers and final consumers. According to the authors, these types of low-temperature PCM microencapsulations may be difficult to achieve via thermally induced polymerization. This was because the temperature-sensitive PCM may become denatured when the operating condition during the microencapsulation process exceeds 40 °C. An effect of this is the resulting changes in the thermophysical properties of the PCM. The mass-loss analysis of the developed mPCMs stabilized within the first eight days of continuous heating in a temperature-controlled oven set at 40 °C after losing only 0.6% of its initial weight. The capsules had an average latent heat of 131.1 kJ/kg and a peak melting temperature of about 8 °C. Notable low-melting PCMs suitable for cold storage application include methyl laurate, tetradecane, tetrahydrofuran, pentadecane, Microtek PCM 6, Sasol Parafol 14–97, PureTemp PT-series (PT4, PT6, PT7, PT8, and PT15), and Rubitherm® RT-series (RT4, RT6, RT12, and RT15).

The use of UV radiation also benefits primarily from the possibility of being performed at low temperatures and can therefore be used to encapsulate heat-sensitive PCMs. It is noteworthy to mention that the encapsulation efficiency and monomer conversion reported are commercially unsatisfactory. Therefore, further research should focus on approaches to enhance process efficiency through selecting more appropriate photoinitiators and using an energy efficient UV lamp, e.g., an LED lamp.

### 2.1.4. Emulsion Polymerization

Emulsion polymerization is similar to suspension polymerization, except the monomer with a surfactant combined in the oil phase, while the initiator dis-solved in the water phase, using a water-soluble initiator. Figure 11 shows the schematic representation of the emulsion polymerization mechanism. In this approach, monomers are first dispersed in the aqueous phase, which leads to the generation of initiator radicals. These radicals migrate into the soaplike micelles swollen with monomer molecules. As the polymerization proceeds, more monomers migrate into the micelle to allow the polymerization to proceed. Because only one free radical is present in the micelle prior to termination, very high molecular weights, on the order of 1,000,000 or higher, are possible. An unpopular emulsion technique, referred to as the inverse emulsion polymerization process, involves dispersing an aqueous solution of monomer in the nonaqueous phase. Typical raw metatrails used in emulsion polymerization are summarized in Table 2.

The most common polymers used for microencapsulation of PCMs via emulsion polymerization are PMMA [114,115], polystyrene [116]. poly(styrene-co-ethylacrylate) [117], n-nonadecane-vinyl copolymer [118], and poly(methyl methacrylate-co-methacrylic acid) [119]. Sarı et al. (2010) [114] synthesized a novel n-heptadecane/polymethylmethacrylate mPCM by means of the emulsion polymerization method. The diameters of mPCMs were found in the narrow range (0.14–0.40 μm) under a stirring speed of 2000 rpm. The DSC analysis shows that the temperatures of melting and latent heats of melting of the PMMA/heptadecane microcapsules were 18.2 °C and 84.2 J/g, respectively. The microencapsulation ratio of the heptadecane in the PMMA microcapsules was found as 38 wt.%. Further, Sarı et al. (2014) [48] prepared polystyrene-coated microcapsules containing capric, lauric, and myristic acids by using emulsion polymerization. The mPCMs melting temperature range of 22–48 °C and a latent heat in range of 87–98 J/g were reported. The mPCMs had a good thermal durability with a PCM retention rate of 95.6% after 5000 cycles.

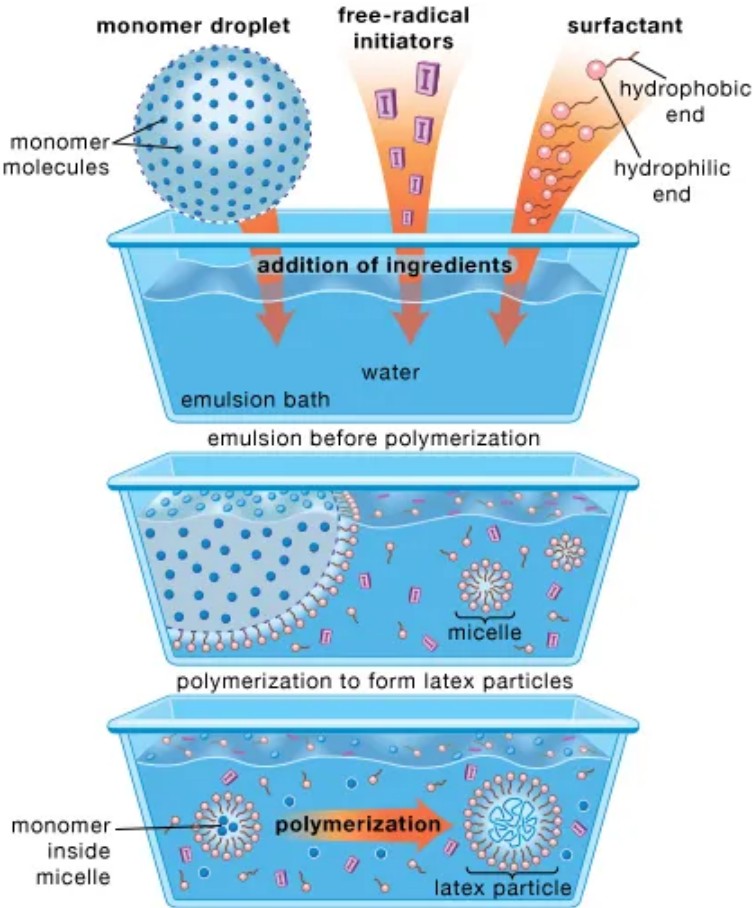

**Figure 11.** Schematic diagram of the emulsion polymerization. Reused from Encyclopædia Britannica, Inc.. Reprinted/adapted with permission from [113]. 2020, Encyclopædia Britannica.

**Table 2.** Raw materials selection for emulsion polymerization.

| Materials | Examples | Consideration and Comments |
|---|---|---|
| Monomers | - Methyl methacrylate<br>- Butyl methacrylate<br>- Styrene<br>- Vinyl esters<br>- Methacrylic acid<br>- Acrylic acid<br>- Hydroxyethyl acrylate<br>- Hydroxyethyl methacrylate<br>- Vinyl acetate | - Monomers are nonreactive with water<br>- Monomers are well suited for free-radical chain polymerization<br>- Monomers with partial water solubility<br>- Carboxyl functional monomers utilize in low levels for particle stability<br>- Hydroxyl functional monomers are reactive with crosslinker<br>- Monomer compatibility and rate of polymerization |
| Initiators | - Sodium, ammonium, or potassium persulfate<br>- Azobisisobutryl nitrile (AIBN)<br>- t-butyl hydroperoxide (TBHP) | - Water soluble for REDOX type initiators (e.g., persulfates)<br>- Thermal initiation for free-radical types (e.g., AIBN, TBHP) |

**Table 2.** *Cont.*

| Materials | Examples | Consideration and Comments |
|---|---|---|
| Surfactants | - Anionic and nonionic types<br>- Anionic types<br>- Sodium lauryl sulfonate<br>- Nonionic types<br>- Nonylphenol ethoxylates | - Limited water solubility<br>- Toxicity issues with nonylphenol ethoxylates |
| Other recipes | - Water<br>- Buffer (e.g., sodium bicarbonate)<br>- Amine<br>- Coalescent<br>- PCMs | - Normally deionized $H_2O$ to minimize variability<br>- Buffer to regulate pH<br>- pH control, stability<br>- Aids film formation<br>- Active energy storage materials |

The production of miniemulsions results when monomers are dispersed in water by means of strong mechanical stirring using a mixed emulsifier system, with a classical emulsifier and a water-insoluble cosurfactant, such as a long-chain fatty alcohol or alkane (e.g., cetyl alcohol or hexadecane). The resulting capsules possess about the same size as the starting monomer droplets. Additionally, the nanoscale particle size capsules have a distribution broader than those obtained by emulsion polymerization. Karaipekli et al. (2019) [49] prepared nanoencapsulated PCMs, where n-octadecane was used as PCM and poly(styrene-co-methacrylic acid) as shell materials by means of the miniemulsion polymerization method. The phase change temperatures of the nanoencapsulated PCMs were in the range of 32.16–32.42 °C and the latent heats were measured as 89.12 J/g and 87.42 J/g for melting and freezing, respectively. The encapsulation ratio of n-nonadecane was determined as 54 wt.%. Şahan et al. (2019) [120] synthesized capsules using SA as the PCM, which was encapsulated in PMMA and four PMMA-hybrid shell materials. The nanoencapsulation was accomplished by miniemulsion polymerization. The mean diameter and the thickness of the spherical shells varied over relatively narrow ranges of 110–360 nm and 17–60 nm, respectively. The variance was indicative of the functional groups of the shell material.

In conclusion, synthesizing mPCMs using chemical methods is void of complications and can be accomplished using the simple polymerization technique. For example, condensation polymerization reactions (in situ and interfacial polymerization) can produce uniform morphological coating with high encapsulation efficiency. However, high skill is needed for preparation due to the nature of the materials' toxicity. Moreover, the formation of polymer shell with a lower molecular weight will result in a weak shell strength and high wall permeability. In contrast, free radical polymerization reactions (suspension and emulsion polymerization) lead to the formation of higher molecular weight shells and thus higher mechanical strength but lower encapsulation efficiency. The summary of thermophysical properties of prepared mPCMs using chemical methods is reported in Table 3.

**Table 3.** Summary of some mPCMs using chemical methods.

| | Core | Shell | $T_{mp}$ (°C) | PS (µm or/nm) | $LH_m$ (J/g) | CC (wt.%) | Ref. |
|---|---|---|---|---|---|---|---|
| In situ polymerization | Paraffin | MUF | 32.4 | $3.02 \pm 0.42$ µm | 134.3 | 77.1 | [38] |
| | n-octadecane | MF | 25.17 | 3 µm | 185.1 | 84.3 | [80] |
| | n-octadecane | UMF | 25.2 | 0.2–5.6 µm | 172.7–190.6 | 71.4–78.8 | [73] |
| | pentadecane | MUF | 12.5 | n/a | 84.5−88.2 | 48.5–50.60 | [75] |
| | capric acid | UF | 33.9 | 0.28 µm | 88 | 78.6 | [121] |
| | n-Tetracosane | MF | 54.2 | n/a | 134.7 | 72.4 | [122] |
| | n-dodecanol | GO-modified MF | 26.3 | n/a | 125.2 | 64.8 | [123] |
| | paraffin | H-SiC-modified MF | 52.3 | n/a | 93.2 | 65.1 | [124] |
| | 1-Dodecanol | MPF | 22.9 | n/a | 169.5 | 88.6 | [41] |
| | oleic acid | $Ag_2O$-UF | 5.4 | | 71.7 | 54.8 | [125] |
| | coconut oil | MF | 21.1 | n/a | 81.9 | 76.2 | [126] |
| | capric acid | nano-SiC-modified MUF | 32.9 | n/a | 97.8 | 65.7 | [127] |
| | 1-dodecanol | MF | 22.1 | 490.2 nm | 79.5 | 40.9 | [39] |
| | Caprylic acid | UF | 19.3 | 200 nm–1.5 µm | 93.9 | 59 | [79] |
| | Paraffin | nanoplatelets laden/UF | 63.2 | 60–65 µm | 110.7–116.7 | 52.2–55.1 | [128] |
| Interfacial polymerization | methyl laurate | CNCs-urea–urethane | $5.48 \pm 0.05$ | n/a | 148.4 | 66.1 | [92] |
| | n-octadecane | PU | 25.2 | 10–14 µm | 220.1 | 71 | [129] |
| | butyl stearate | PU | 23.2 | 1–5 µm | 85 | 69.7 | [130] |
| | dodecanol dodecanoate | PU | 31.2 | 10 to 40 µm | 103.4–140.3 | 54–74.5 | [42] |
| | butyl stearate | PU | 23.2 | - | 80.6 | 74.3 | [89] |
| | Methyl laurate | nano-$TiO_2$-PU | 7.4 | 150–350 µm | 147.71 | 83.3 | [131] |
| | paraffin wax | PU | | 25–185 nm | 153.9 | 80.2 | [91] |
| | n–octadecane | $Fe_3O_4$-PU | 29.34 | 35–500 µm | 165.7 | 83.6 | [68] |
| Suspension polymerization | n-hexadecane | $Fe_3O_4$-modified PMMA | 18 | 180 µm | 53.1 | 24.4 | [132] |
| | Paraffin@RT21 | Cross-linked PMMA | 21 | 5–10 µm | 113.4 | 85.6 | [43] |
| | n-hexadecane | PMMA or/ poly(BA-co-MMA | 18.3 | n/a | 63.1 | 28.9 | [33] |
| | Paraffin@RT21 | Cross-linked PMMA | 21 | n/a | 93.1 | 66.5 | [45] |
| | Paraffin wax | Poly (DVB/St/AA) | 56.1 | 200–500 µm | 62.4 | 41.1 | [133] |
| | n-octadecane | PDVB | 30 | n/a | 184.0 | 76.3 | [98] |
| Emulsion/miniemulsion polymerization | Paraffin | PMMA | 29.8 | n/a | 75.6 | 72.5 | [134] |
| | heptadecane | PMMA | 20.3 | 0.14–0.40 µm | 81.5 | 38 | [114] |
| | n-heptadecane | PSt | 21.4 | 1–15 µm | 136.9 | 63.3 | [135] |
| | n-octadecane | Poly(St-MMA) | 30.9 | $102 \pm 11$ nm | 117.3 | 49.0 | [136] |
| | n-octadecane | $Cu_2O$/PMMA | 27.2 | n/a | 102.66 | 70.0 | [137] |

$T_{mp}$: Peak melting temperature (°C), PS: particle size (µm or/nm), LHm: latent heat of melting (J/g), CC: encapsulated PCM content (%) and n/a: not available

### 2.2. Microencapsulation of PCMs with Other Methods and Shell Types

2.2.1. Coacervation-Phase Separation

The coacervation-phase separation is a microencapsulation technique that involves a physicochemical approach for the preparation of mPCM. It comprises two oppositely charged polyelectrolytes (polycation and polyanion) in an aqueous solution. The polycation is typically gelatine, while the polyanion is Arabic gum. Tiny droplets of the oil phase

(PCMs) are dispersed in the aqueous phase containing polycation as an emulsion. The polyanion solution is then added proportionately to the formed emulsion with moderate agitation. Upon lowering the pH system using acid, phase separation is induced, and a polymer-rich phase (coacervate) is formed and deposits on the oil-interface droplets to form a gelatinous shell on cooling. The shell can then be cross-linked using glutaraldehyde, which hardens and prevents the gelatine from melting during PCM phase transitions. The encapsulation progression based on coacervation-phase separation is illustrated in Figure 12.

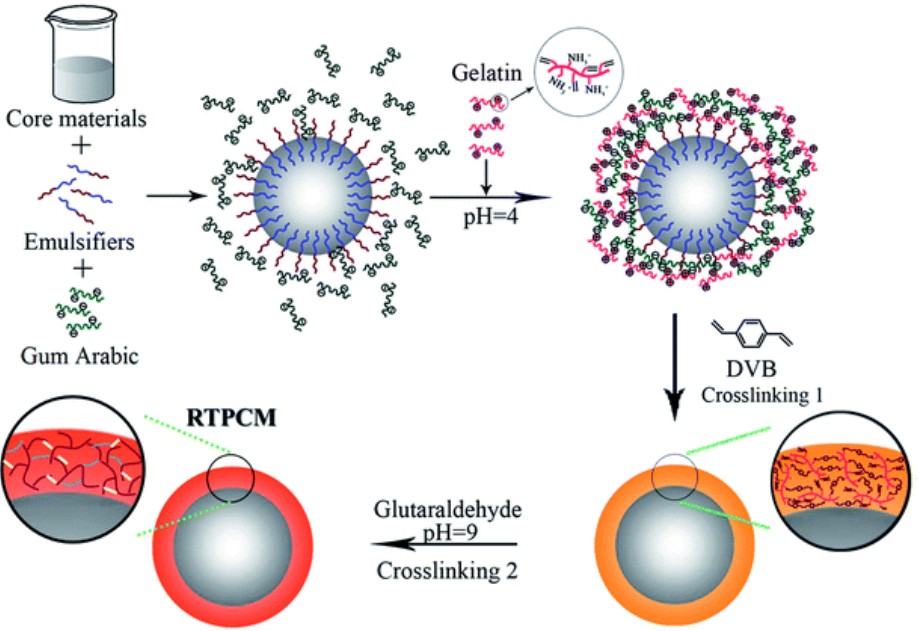

Core materials: 2-Phenylamino-3-methyl-6-di-n-butylamino-fluoran, 2,2-bis(4-hydroxyphenyl) propane, 1-hexadecanol,

Emulsifiers: Tween 20 + Span 80

**Figure 12.** Synthesis routes of preparation of mPCMs by means of coacervation-phase separation. Reused from Wu et al. (2017) [138].

Several works in the literature have surfaced over the last two decades on PCM microencapsulation using coacervation-phase separation [50,139,140]. Wu et al. (2017) [138] prepared a novel mPCM by complex coacervation. The ingredients used include modified gelatin containing vinyl groups and gum Arabic. The approach uses a spironolactone derivative color former-phenolic hydroxyl compound color developer and 1-hexadecanol, respectively, being the cosolvent and PCM. The impact of several key parameters such as the substitution degree of vinyl groups on modified gelatin, the type and amount of divinyl cross-linkers, and the core/shell ratio on properties of the mPCMs were investigated. Stable microcapsules (of uniform particle diameters in the range of 7–10 μm) with latent heat of about 72 kJ/kg and encapsulation efficiency >85% were produced. The as produced mPCMs possess good thermal reliability after 100 thermal cycles. An obvious change in color was observed after multiple thermal cycles were performed. As shown in Figure 13, the dispersion was white at 60 °C, indicating fully charged PCM, whereas the white color turned gray at 15 °C, signaling completely discharged PCM. There was no color change after 100 thermal cycles.

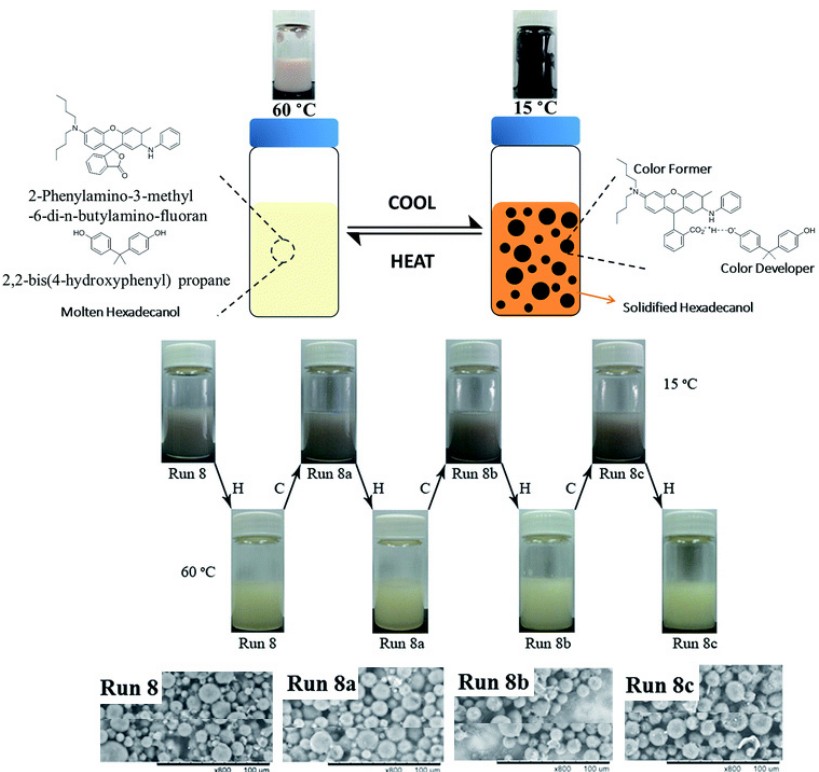

**Figure 13.** Color change images of Run 8 microcapsules, respectively, at 15 °C (gray) and 60 °C (white) in multiple thermal cycles with corresponding SEM images. Run 8 (25 cycles), Run 8a (50 cycles), Run 8b (75 cycles), and Run 8c (100 cycles). Reused from Wu et al. (2017) [138].

Ma et al. (2021) [141] developed a new phase separation method for the microencapsulation of oxalic acid dihydrate/boric acid eutectic (OA-PCM) via inducing the coacervation of ethyl cellulose (EC) and acrylonitrile-butadiene-styrene (ABS) to form a hybrid shell with the addition of polydimethylsiloxane. The effect of EC types, EC:ABS ratios, and different core/shell ratios on mPCMs properties were investigated. The mPCMs achieved at an EC:ABS:OA-PCM ratio of 1:1:2 displayed an optimal performance with a latent heat of 178.4 kJ/kg and melting temperature of about 78 °C. Although the microencapsulation process which takes place at ambient temperature was adjudged to be simple and eco-friendly, the materials used to encapsulate PCM via means of complex coacervation method is a host for bacteria growth; hence, it is not commonly used in building and medical applications.

### 2.2.2. Spay Drying

Spray drying is a suitable microencapsulation technique for containing heat-sensitive materials and has been widely used in the food [142] and pharmaceutical industries [143], as well as for PCMs [63]. Spray drying is a promising technique when utilized on an industrial scale as it offers lower production costs, limited loss of raw materials and process waste, and ease of control and potential scale up to a large, continuous process [144]. Methaapanon et al. (2020) [62] prepared mPCMs with silica shell matrices using the spray drying method. The silica solid shell was formed through polycondensation during the spray drying of the PCM/sol–gel emulsion. Hawlader et al. (2003) [139] prepared microcapsules using the spray drying technique. In their study, gelatin-acacia was used as the polymer shell and paraffin wax as PCM. The microcapsules possess high core loading up to 80% with an average particle size of 0.15 μm. Depending on the core-to-coating ratio, the encapsulation efficiency was in the range of 60–92%. Similar techniques involving the use of low-density polyethylene (LDPE) and ethyl vinyl acetate (EVA) copolymer to form shells were reported [63].

The synthesis of mPCMs using the spray drying process consists of the following:

(1)   Homogeneous liquid solution (feed stream) preparation—consists of phase change material and dis-solved polymer, which is achieved through the use of a proper solvent;

(2)   Atomization of the as-prepared solution by means of a carrier gas stream (such as compressed nitrogen);

(3)   Solvent evaporation, where the particles were dried by hot nitrogen stream (dried nitrogen) in the drying chamber, and then the final product was recovered in the collector, as shown in Figure 14.

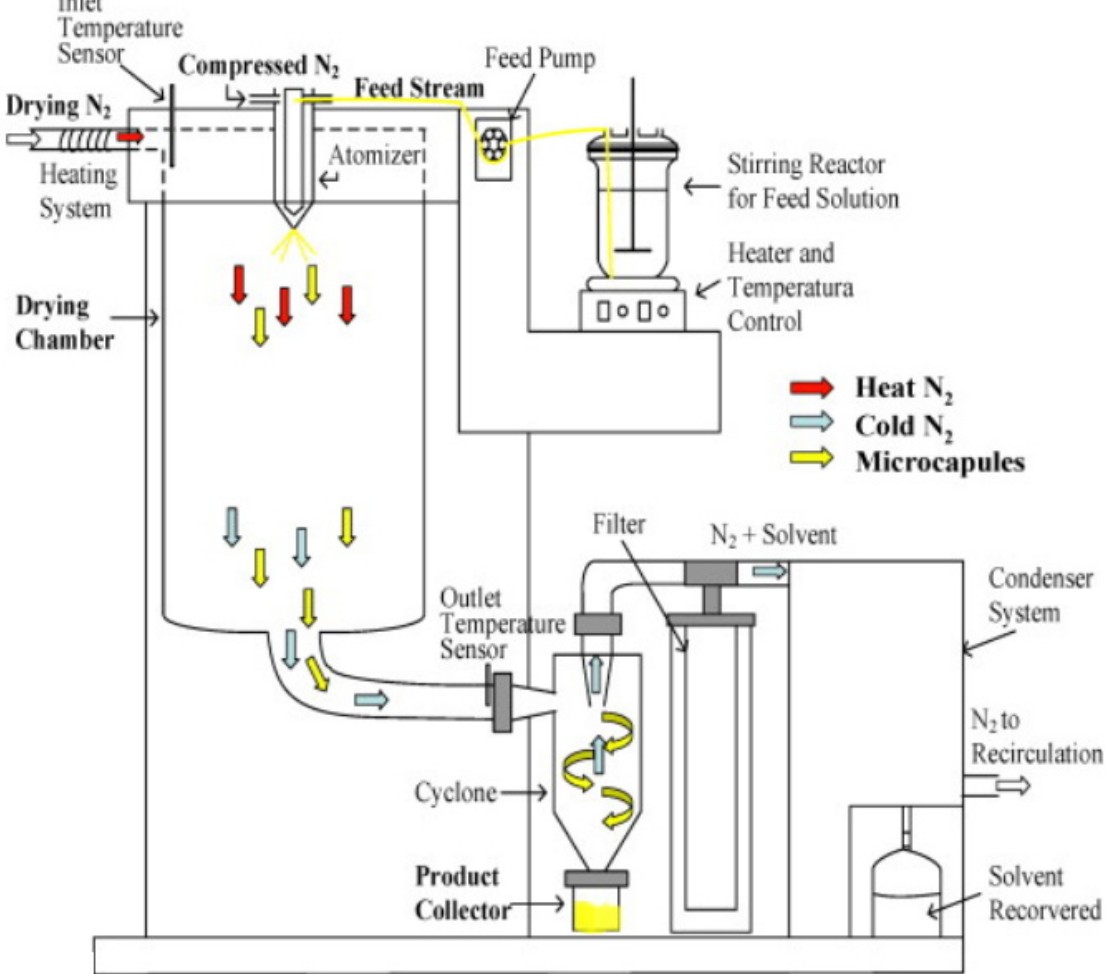

**Figure 14.** Schematic representation of the spray drying equipment used for fabricated mPCMs. Reused from Borreguero et al. (2011) [63].

In general, the mPCMs produced using the spray dryer technique face the problem of PCM leaking out of the shell due to the formation of pores.

As noted earlier, the use of polymeric shells in the microencapsulation of PCMs are common, owing to their ease of processing, excellent sealing characteristics, structure flexibility, and chemical and thermal stability. Nevertheless, their use is limited because of the polymer susceptibility to flame and poor thermal conductivity issues [145]. Therefore, investigating alternative shell materials that possess good flame retardancy, high thermal conductivity, and rigidity, such as inorganic shells, is required. In the following subsections, the common methods used to encapsulate PCMs with inorganic shell are discussed.

### 2.2.3. Sol–Gel Method

This method is classified under the physical–chemical category, and the common shell materials used are silica [146] and titanium oxide [147]. It attracts more attention from researchers due to its mild processing conditions. Figure 15 shows the schematic route of mPCMs' formation using the sol–gel approach with silica shell and n-octadecane as the core. Sodium silicate served as the precursor. The production process consists of three steps, which are:

(1) The formation of PCM O/W emulsion through the mixing of PCM with a surface-active solution containing surfactant (emulsifier);

(2) The aqueous acidic phase (sol solution) prepared by dis-solving the precursor compound, e.g., tetraethyl orthosilicate (TEOS) or sodium silicate precursor in water;

(3) Microcapsules formation via condensation polymerization by dropwise addition of the sol solution into the PCM O/W emulsion.

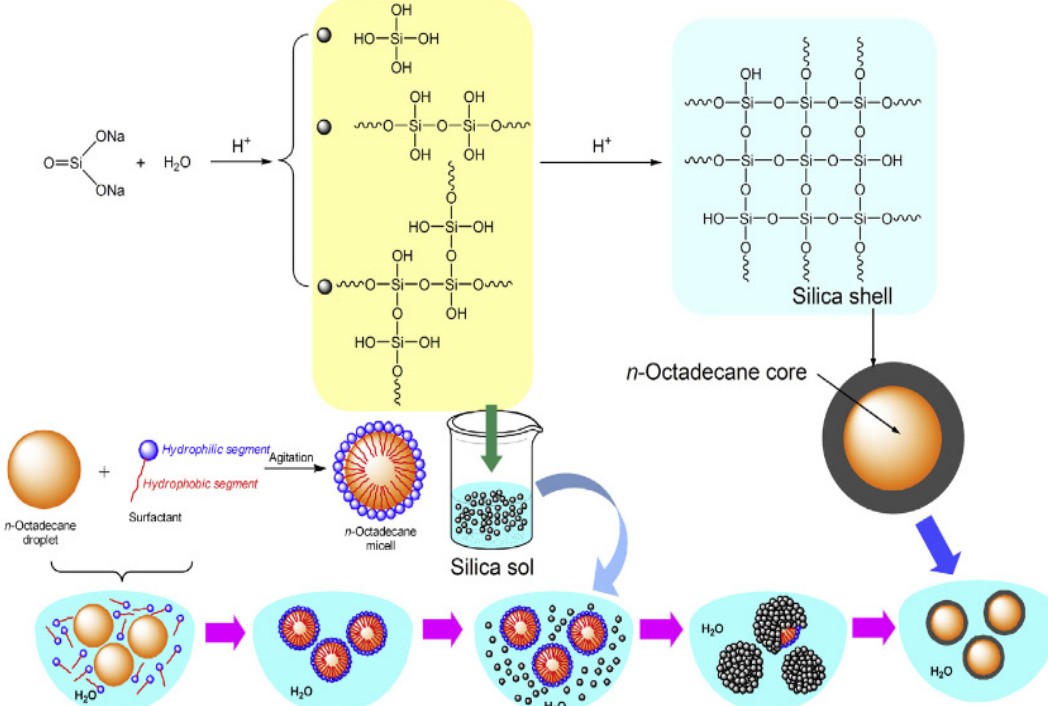

**Figure 15.** Schematic representation of the formation mechanism for the microencapsulated n-octadecane with silica shell material with sodium silicate precursor. Reused from He et al. (2014) [148].

TEOS and Tetra-n-butyl titanate (TNBT) are the mostly used precursors for producing mPCMs with $SiO_2$ and $TiO_2$ shells, respectively [149–153]. Ji et al. (2021) [58] prepared composite $GO/TiO_2$ paraffin microcapsules by interfacial condensation polymerization in a sol–gel system. The composite appeared to have a spherical core-shell structural morphology, and the GO nanosheets self-assemble on the surface of the microcapsules by sharing electrons and hydrogen bonds. The latent heat of microcapsules was more than 160.75 J/g, and the thermal conductivity enhanced from 0.195 to 0.297 W/(m·K). Jin et al. (2017) [154] investigated the effect of acidic pH on the encapsulation efficiency of n-eicosane with different inorganic silica precursors, including TEOS and sodium silicate. According to the authors, mPCMs synthesized at pH of 2.20~2.30 resulted in the formulation of nanoencapsulated n-eicosane and displayed spherical shapes of excellent phase change properties and high encapsulation efficiency. Overall, the n-eicosane/sodium silicate nanocapsules presented good properties and thermal stability when prepared at pH 2.90~3.00. However, pH lower than 2.9 resulted in very poor encapsulation efficiency and low enthalpies. Later on, Yuan et al. (2019) [146] studied the influence of alkaline pH on

the formation of lauric acid/SiO2 nanocapsules via the sol–gel process. The results showed that the pH range of 9.4–10.2 is suitable for the synthesis of nanocapsules with high latent heat (160.0 J/g) and small particle size (357 nm). Figure 16a shows the TEM images of two samples prepared at different alkaline pH values, sample-a (pH = 10.4) and sample-b (pH = 12.4). As can be seen in Figure 16a, the dark core is surrounded by a pale shell for sample-a, which confirmed the core-shell structure. However, for sample-b, the particles show a uniform dark color where the core–shell structure cannot be found, which signifies the formation of nanoparticles with empty PCM. The same conclusion was confirmed by EDS analysis (Figure 16b). In sample-a, carbon with the an atomic percentage of 77.9% is the main element in the nanocapsules, while silicon and oxygen atoms account for a minor percentage. In sample-b, the atomic percentage of oxygen and silicon are dominated with 72.1% and 24.8%, respectively, and small carbon accounts, which means that solid nanospheres with almost empty PCM were formed. The DSC measurements also confirmed the results obtained by TEM and DSC analysis (Figure 16c). Li et al. (2018) [155] confirmed the same results obtained by Yuan et al. (2019) [146], where the overall thermal properties of TiO$_2$ mPCMs were enhanced under acidic conditions.

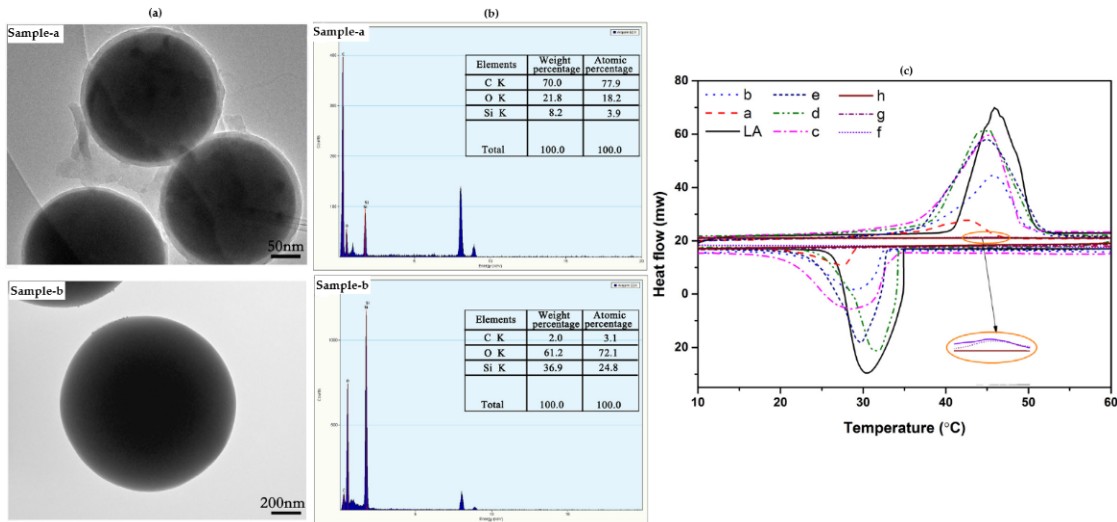

**Figure 16.** (**a**) TEM images of the samples synthesized at various values of pH: sample-a (pH 10.4) and sample-b (pH 12.4), (**b**) corresponding DSE analysis for sample-a and sample-b, and (**c**) DSC measurements. Reused from Yuan et al. (2019) [146].

The encapsulation of PCM with shells such as TEOS and TNBT are relatively expensive and unattractive, especially when considered for use at the lab scale. To this end, He et al. (2014) [148] examined the manufacture of PCM microcapsules via the sol–gel method utilizing a sodium silicate precursor, which is cost effective. At a controlled pH of 2.95–3.05, spherical PCM microcapsules were prepared. Nevertheless, the mPCMs showed a core/shell mass ratio and microcapsule efficiency, respectively, of 41.8 wt.% and 41.5%. A few years later, Pourmohamadian et al. (2017) [156] microencapsulated PA with inorganic SiO$_2$ shell via the sol–gel method in alkaline medium via sodium silicate precursor. The optimum pH was found to be pH 11, were the sample had a perfect spherical shape with a smooth surface. Further, DSC measurements showed that the as-prepared mPCMs has similar phase change behaviors as those of pure PA PCM, which melts at 67.2 °C, freezes at 56.5 °C, and possesses an average latent heat of 107.2 kJ/kg. Zhang et al. (2021) [157] successfully prepared mPCMs through in situ dehydration and condensation reaction. The microcapsules consist of silica shell derived from sodium silicate and paraffin wax (PW) forming the core. The mPCMs with core to shell mass ratio of 4:1 showed interesting results in terms of energy storage capacity. Following that, the DSC, TGA, and thermal reliability tests demonstrated that the prepared mPCMs possessed depressed supercooling,

good thermal stability, and long-term thermal stability against the heating/cooling cycling (Figure 17). The low cost and availability of sodium silicate and PW makes the encapsulation technology proposed in this work have great potential for practical application in the field of thermal management, including packaging, clothing, and architecture.

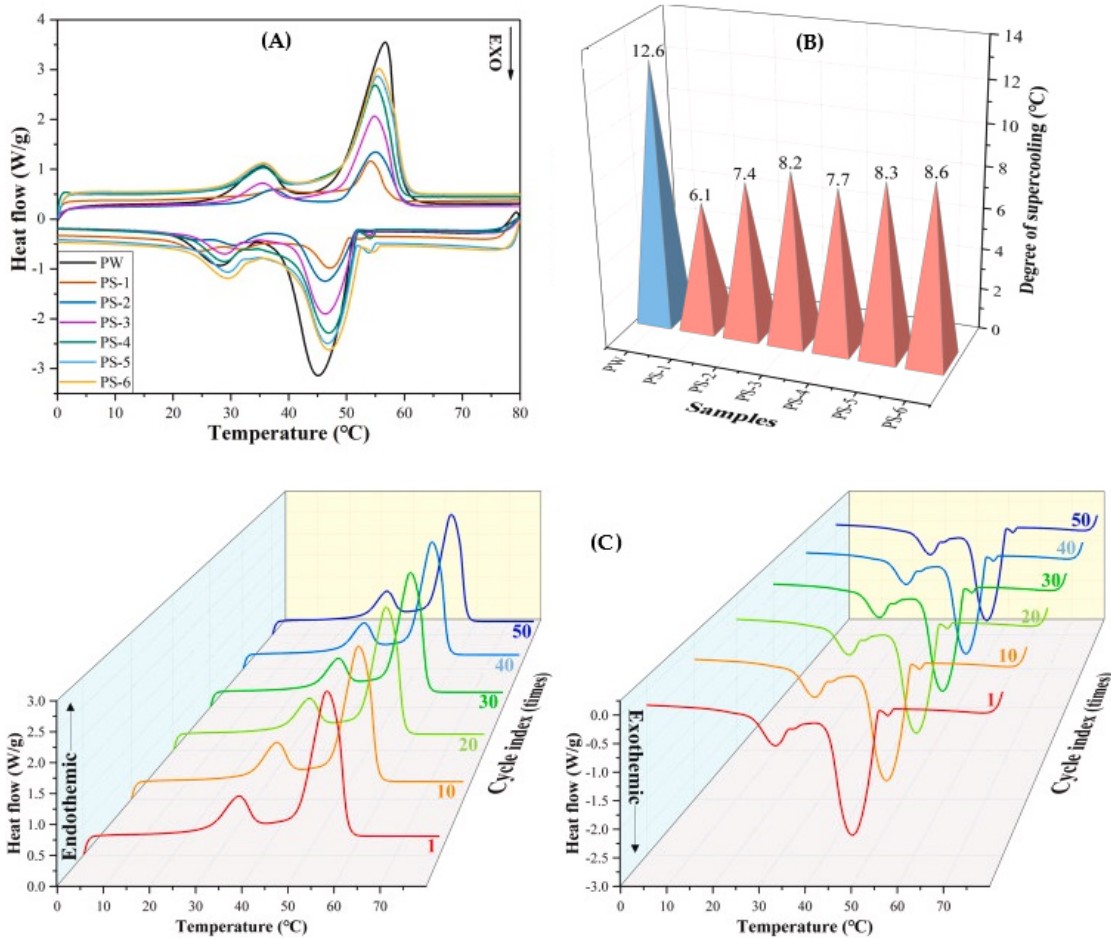

**Figure 17.** The as-prepared mPCMs, (**A**) DSC curves, (**B**) degree of supercooling, and (**C**) DSC curves of PS-5 under 50 loops of the heating–cooling cycle. Reused from Zhang et al. (2021) [157].

## 2.2.4. Self-Assembly Method

Thermal performance, stability, and reliability are critical parameters that are carefully considered in the production of PCM microcapsules. Studies have shown that although PCM microcapsules synthesized with $SiO_2$ shell may display improved thermal performance, their shells are not robust enough to withstand harsh conditions; hence, the microcapsules are easily damaged. Furthermore, the TEOS precursor is not cheap when considered for upscaled production of PCM microcapsules. Consequently, finding an inexpensive and easy to process alternative inorganic shell is of great interest to many researchers. There is no doubt that calcium carbonate ($CaCO_3$) has the properties of low cost, higher thermal conductivity, high rigidity, and good thermal and chemical stabilities compared to synthetic polymer [145]. Yu et al. (2014) [61] microencapsulated n-octadecane PCM using $CaCO_3$ as the shell material through a self-assembly technique. Figure 18 illustrates the schematic of the formation mechanism for the microencapsulated n-octadecane with $CaCO_3$ shell via a self-assembly approach. At first, the PCM O/W emulsion is formed by mixing PCM with a surfactant, and in this case, a blend of Span 80 and Tween 80 surfactants was utilized. Following that, the $CaCl_2$ aqueous solution was added dropwise into the emulsion system, where the $Ca^{2+}$ was assembled on the surface of the PCM micelles through the complexation taking place between $Ca^{2+}$ and hydroxyl groups

of surfactants. Finally, an aqueous solution of Na$_2$CO$_3$ was introduced to the emulsion and the precipitation reaction between Ca$^{2+}$ and CO$_3$$^{2-}$ was observed to be responsible for the formation of the calcium carbonate shells. PCM microcapsules with core/shell structure were attained with an excellent spherical morphology. The thermal conductivity, durability, and stability of the PCM microcapsules were improved in comparison to those of the SiO$_2$ shell. Additionally, the CaCO$_3$ shell enhances the n-octadecane crystallinity due to the induced α-form crystallization by heterogeneous nucleation.

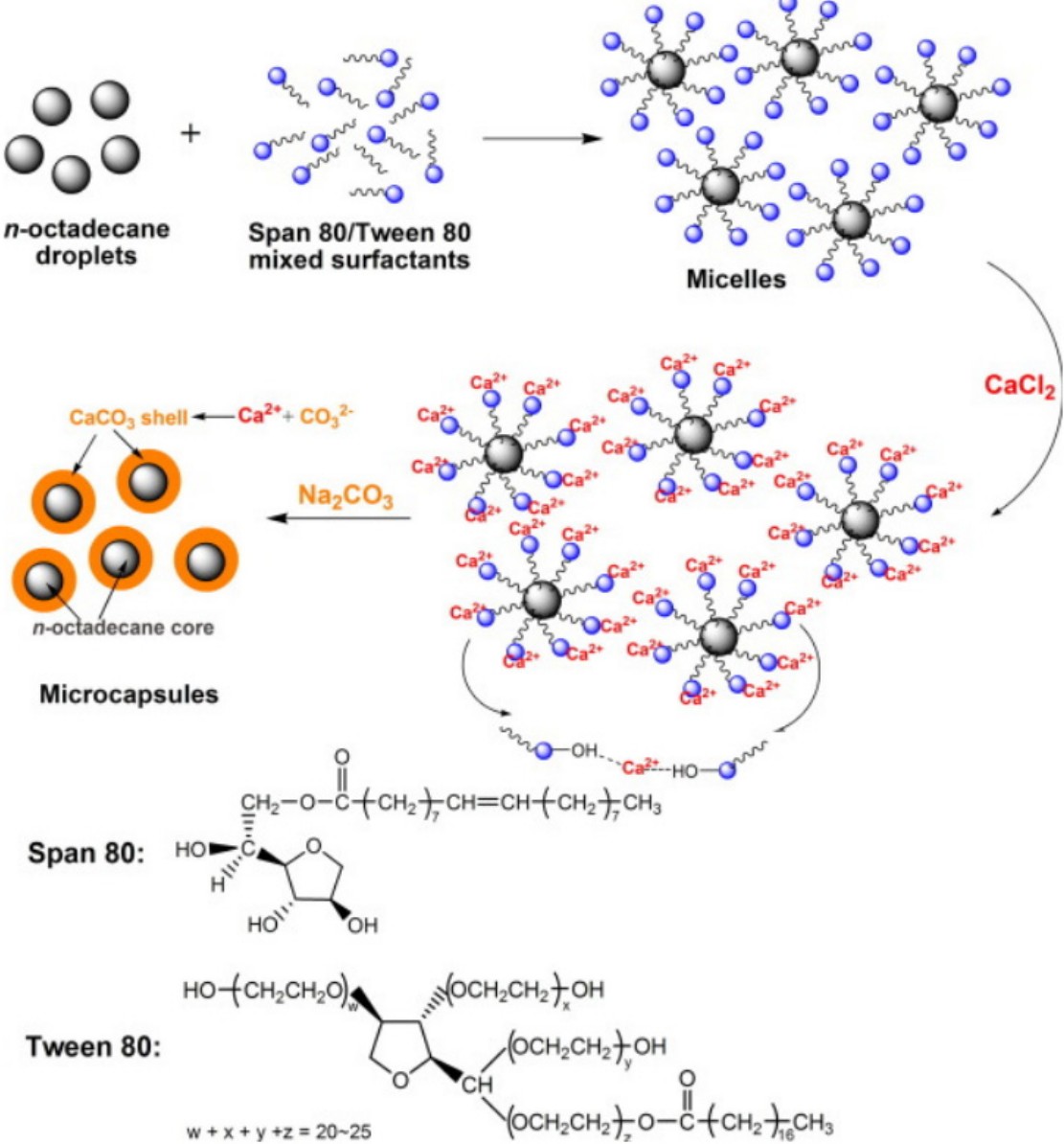

**Figure 18.** Scheme of formation mechanism for the microencapsulated n-octadecane with CaCO$_3$ shell via a self-assembly method. Reused from Yu et al. (2014) [61].

Several articles have been published on the use of CaCO$_3$ as a shell material for microencapsulation of PCMs via the self-assembly method [158–160]. Sari et al. (2021) [161] synthesized mPCMs with calcium carbonate as a shell and heptadecane (HD) as a core via the self-assembly method. The DSC results revealed that the produced HD/CaCO$_3$ microcapsules had a melting temperature of around 16 °C, and the latent heat capacity ranged from 85.6 to 147.3 J/g. The cycled microcapsules showed excellent chemical stability and remarkable TES dependability with the 1000-cycling treatment. The microcapsules

had outstanding thermal durability, as demonstrated by their thermal degradation over the phase change temperature of HD. Jiang et al. (2018) [162] synthesized modified PCM microcapsules with $CaCO_3$ and graphene oxide (GO) shells for improved thermal energy storage and PCM exudation problems. Excellent spherical core-to-shell structures with an encapsulation ratio of 73.2%, thermal conductivity of 0.86 W/(m·K), and good mechanical properties were obtained when 1.0 wt.% of GO was added to the system. The addition of GO led to crack formation on the $CaCO_3$ shell. Nevertheless, the authors believed that the GO possesses a good barrier property that precludes PCM from leaking. Wang et al. (2021) [163] synthesized mPCM with control morphology using a $CaCO_3$ shell. The microcapsules' apparent morphologies could be altered from spherical to spindle shapes by increasing reaction temperature and surfactant concentrations (Figure 19).

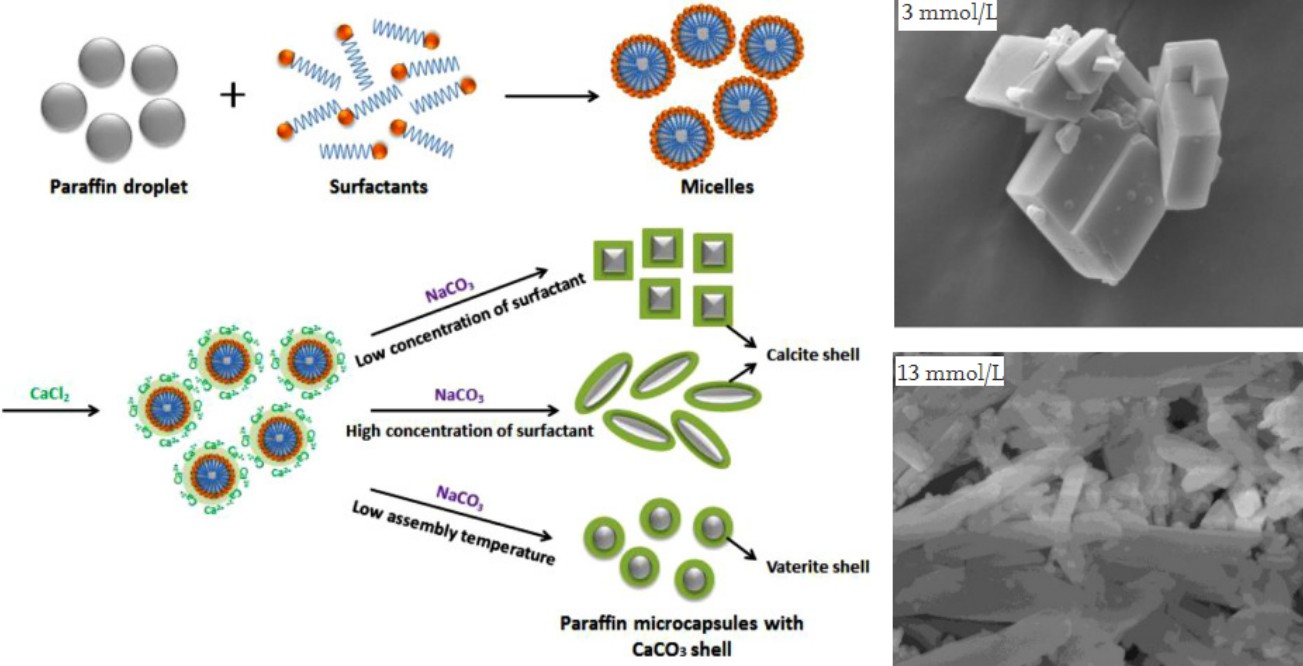

**Figure 19.** Scheme for the preparation of paraffin@$CaCO_3$ microcapsules at different surfactant concentrations. Reused from Wang et al. (2021) [163].

Other shell materials were used to encapsulate PCMs via a self-assembly method. Gao et al. (2017) [164] used cuprous oxide ($Cu_2O$) to encapsulate n-eicosane by using an emulsion-templating self-assembly method along with in situ precipitation. Figure 20 is an illustration of the self-assembly synthetic route and reaction mechanism adopted to produce $Cu_2O$ microcapsule containing n-eicosane. According to the authors, ingredients such as $CuSO_4$ (copper source), CTAB (cationic surfactant), NaOH (precipitation agent), and glycose (reducing agent) were used to form the $Cu_2O$ shell.

The microstructures and morphologies of the as-prepared mPCMs were impacted by the surfactant, alkali concentrations, and copper source. When synthesized at the optimum conditions ($C_{CTAB}$ = 0.15 mol/L, $C_{NaOH}$ = 5.00 mol/L, and 1:1 mass ratio of PCM to $Cu_2O$ shell), the microcapsules displayed an octahedral morphology and typical core–shell structure, as shown in Figure 21. The as-prepared mPCMs possess high encapsulation efficiency and storage capacity, rapid thermal response, as well as an excellent thermal stability and thermal reliability. There was also an increase in the mPCMs' thermal conductivity, coupled with a low degree of supercooling due to the encapsulation of n-eicosane with a highly thermally conductive inorganic wall. The mPCMs showed solar thermal energy-storage capability through solar photothermal conversion and displayed a high solar photocatalytic activity toward organic dyes under sunlight radiation. Furthermore, in

the presence of the $Cu_2O$ shell, the microcapsules showed a gas-sensitive property to some harmful organic gases. tr

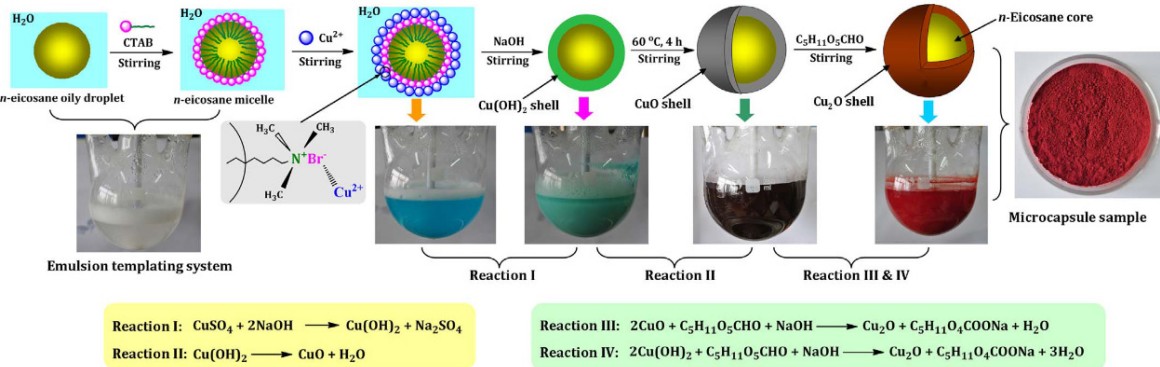

**Figure 20.** Scheme of self-assembly synthetic route and reaction mechanism for the bifunctional microcapsules based on an n-eicosane core and $Cu_2O$ shell. Reused from Gao et al. (2017) [164].

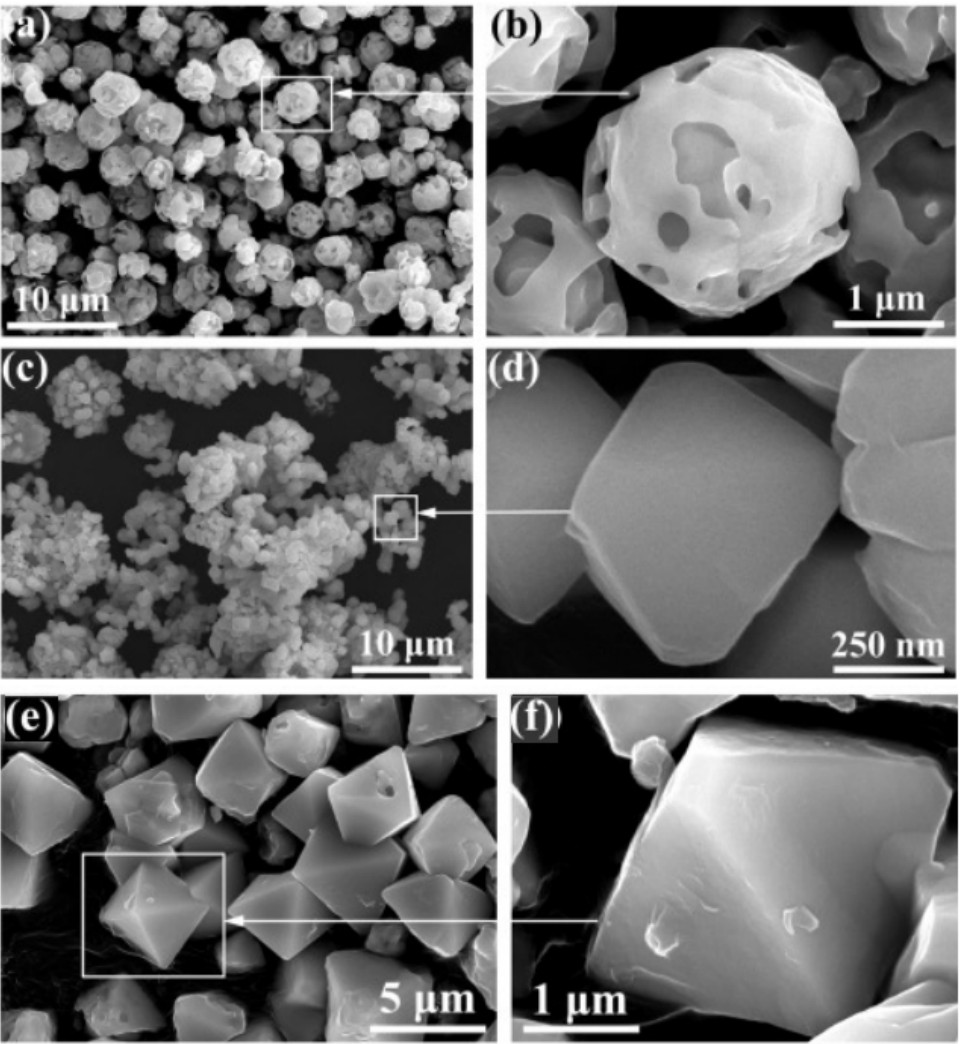

**Figure 21.** SEM micrographs of the mPCMs samples synthesized at fixed surfactant and alkali concentrations of 0.15 mol/L and 5.00 mol/L but at different n-eicosane/$CuSO_4$ mass ratios of (**a,b**) 60/40, (**c,d**) 40/60, and (**e,f**) 50/50. Reused from Gao et al. (2017) [164].

Furthermore, Cia et al. (2022) [165] successfully fabricated paraffin mPCMs (Pn@PWO) with lead tungstate (PbWO$_4$) shell by in situ precipitation and self-assembly methods. The amount of trisodium citrate dihydrate (TSCD) plays significant role in determining the capsules' surface morphology, including a spindle or spherical shape, through control growth of PbWO$_4$ shell. The as-prepared mPCMs possess a high latent heat-storage capacity over 100 J/g, and exhibit high thermal conductivity over 0.596 W/(m·K), and positive leakage-prevention performance. The mass attenuation coefficients of microcapsules at 86.5 keV and 105.3 keV reach 1.98 and 2.08, respectively, showing excellent gamma shielding performance. This type of mPCMs has a potentially wide application, including textiles and nuclear engineering buildings which require simultaneous gamma radiation shielding and thermal regulation.

### 2.2.5. Microfluidic Method

Some unique PCM microencapsulation methods are rarely reported in the literature, such as the microfluidic PCM microencapsulation technique [166,167]. Lone et al. (2013) [68] presented, for the first time, a microfluidic approach toward the fabrication of highly monodisperse polyurea microcapsules containing n-octadecane as the core. According to the study, the mPCMs had a diameter size in the range of 35–500 μm. The synthesis consisted of the following two steps:

(1) Emulsification of n-octadecane, isophorone diisocyanate (IPDI), and dibutyltin dilaurate (DBTDL) in an aqueous mixture of tetraethylenepentamine (TEPA), poly (vinyl alcohol), and sodium dodecyl sulfate (SDS);
(2) In situ polycondensation between TEPA and IPDI along and outside the tube length.

Consequently, through the microfluidic approach with coflowing channels, Fu et al. (2014) [69] were able to prepare elastic silicone/n-hexadecyl bromide microcapsules. According to the authors, the procedure involved the formation of a double oil1-in-oil2-in-water (O1/O2/W) droplet with a core–shell geometry, as shown in Figure 22A,B. The as-prepared mPCMs had a spherical shape with a low PCM content of 49 wt.% (Figure 22C–E). However, the PCM content may increase by adjusting the rate of the three fluids used in the microfluidic system.

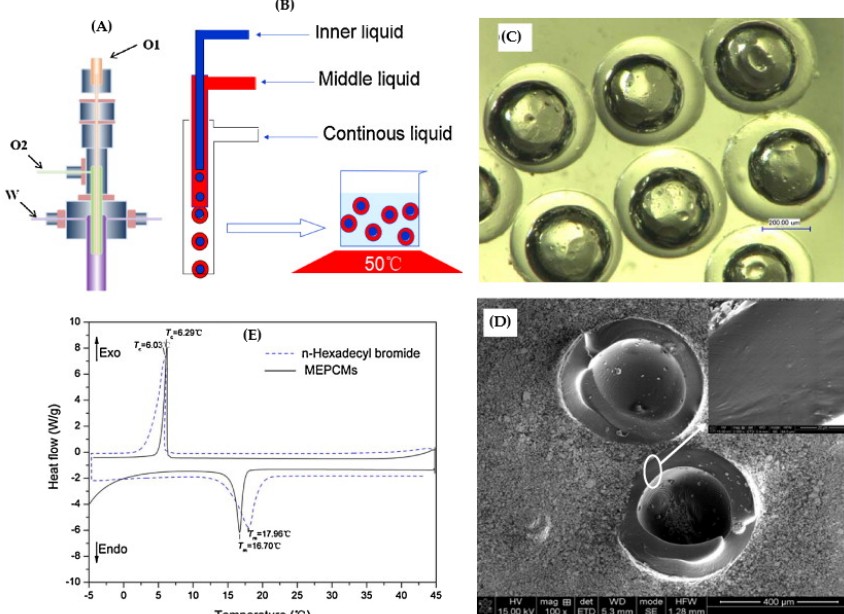

**Figure 22.** (**A**) Illustration of the microfluidic device, (**B**) fabrication process of double emulsions, (**C**) optical microscope of mPCMs, (**D**) cross-sectional scanning electron microscopy image of mPCMs, and (**E**) DSC thermogram of n-hexadecyl bromide and mPCMs. Reused from Fu et al. (2014) [69].

Furthermore, Rahman et al. (2012) [168] produced monodisperse mPCMs using membrane emulsification followed by suspension polymerization. Monodisperse O/W emulsion was prepared with Shirasu Porous Glass hydrophilic membranes. The selected membrane pore sizes are 10, 10.2, and 20 μm, and the polymerization of MMA was carried out in a batch reactor at a temperature range of 70–90 °C. The results showed that the different membranes (10, 10.2, and 20 μm) produced mPCMs with excellent average diameters of 22.4 ± 1.5, 25.4 ± 0.8, and 37.5 ± 1.69 μm, and average latent heats of 113.9 ± 12, 116.7 ± 1.4, and 109.9 ± 8.7 J/g, respectively. However, the mPCMs produced using this method exhibited low mechanical strength in comparison with commercially available mPCMs. Likewise, Akamatsu et al. (2019) [70] encapsulated paraffin's PCM with silicone-based shells, using a glass capillary device. A few years later, Watanabe et al. (2022) [169] adopted the same approach to formulate microcapsules where a monodisperse biocompatible cellulose acetate (CA) constituted the shell materials and HD was used as the core material. Figure 23 shows the schematic representation of the microflow process used to produce the cellulose acetate-n-hexadecane microcapsules. The microcapsules were prepared by combining microfluidic droplet formation and subsequent rapid solvent removal from the droplets by solvent diffusion. The as-prepared mPCMs exhibited latent heat storage 179 J/g, corresponding to 66 wt.% of PCM content. This process contributes to the preparation of environmentally friendly microcapsules for heat storage applications. The same process was used elsewhere [167] for encapsulation paraffin using a copolymer as the shell material. The copolymer used was IPDI and TEPA. The results showed that the microcapsules had an average latent heat of 87.5 J/g with a calculated encapsulation efficiency of 96.5%.

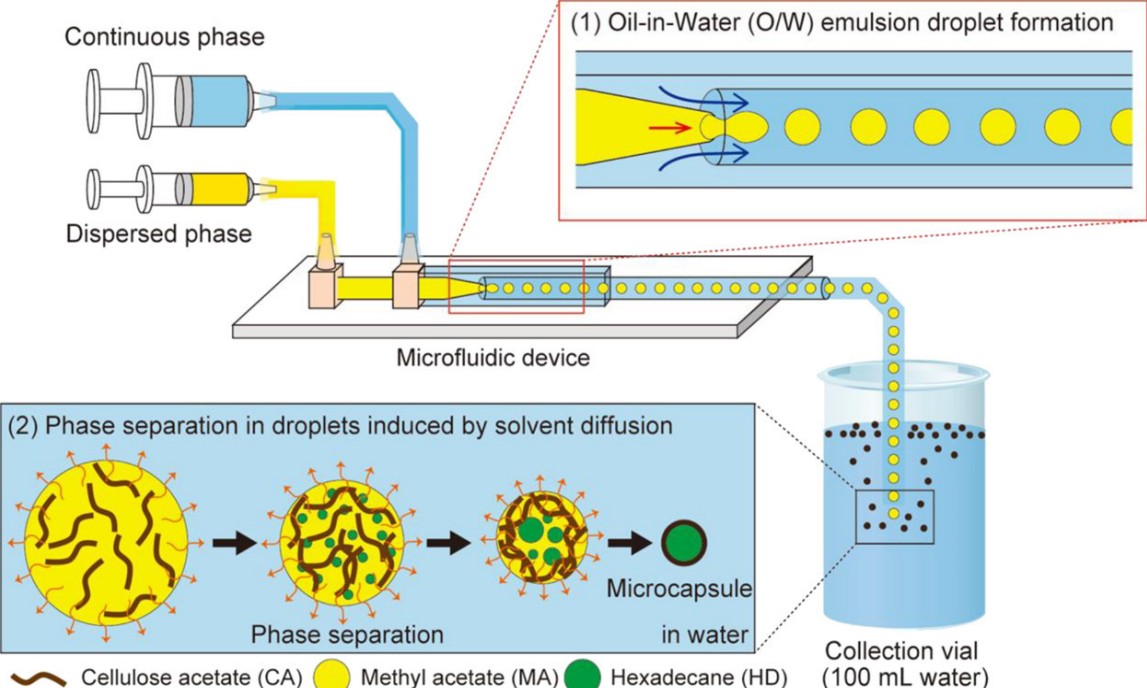

**Figure 23.** Schematic route of a microflow process for the production of cellulose acetate-n-hexadecane microcapsules. Reused from Watanabe et al. (2022) [169].

### 2.2.6. Solvent Evaporation/Phase Separation

The microencapsulation method involving solvent evaporation and phase separation is commonly used for microencapsulating bioactive materials used in the pharmaceutical industry. The approach is consistent with the encapsulation of hydrophobic materials which involves an o/w emulsion system. However, encapsulation of hydrophilic materials such as water-soluble drugs can also be accomplished by double water-in-oil-in-water (w/o/w)

emulsions. A benefit of the double emulsion process is the possibility of encapsulation of hydrophilic cores. Figure 24 shows the scheme representation and experimental steps of mPCMs preparation by means of solvent evaporation, followed by internal polymer phase separation. In this method, the polymer is dis-solved in a core material (PCM) with the help of proper solvent, and then the mixture is added to the water solution which contains surfactant under stirring; thus, oil droplets are obtained. The evaporation of the solvent triggered the internal phase separation in the polymer/PCM/solvent droplets. The phase separation of the polymer-rich part is caused by the interfacial tension interaction between core, polymer, and aqueous phases. Hence, the polarity and interfacial tension of the polymer within the PCM droplets are crucial for forming the core/shell structure mPCMs with complete polymer shell coverage. Figure 25 represents the SEM images of three possible mPCMs' surface morphologies, including (a) core/shell, (b) acorn-shaped, and (c) golf ball-like morphologies. The surface morphology of PMMA capsules is very sensitive to the polarity, interfacial tension, and molecular weight (Mw) of PMMA, emulsifier type, and concentration [170].

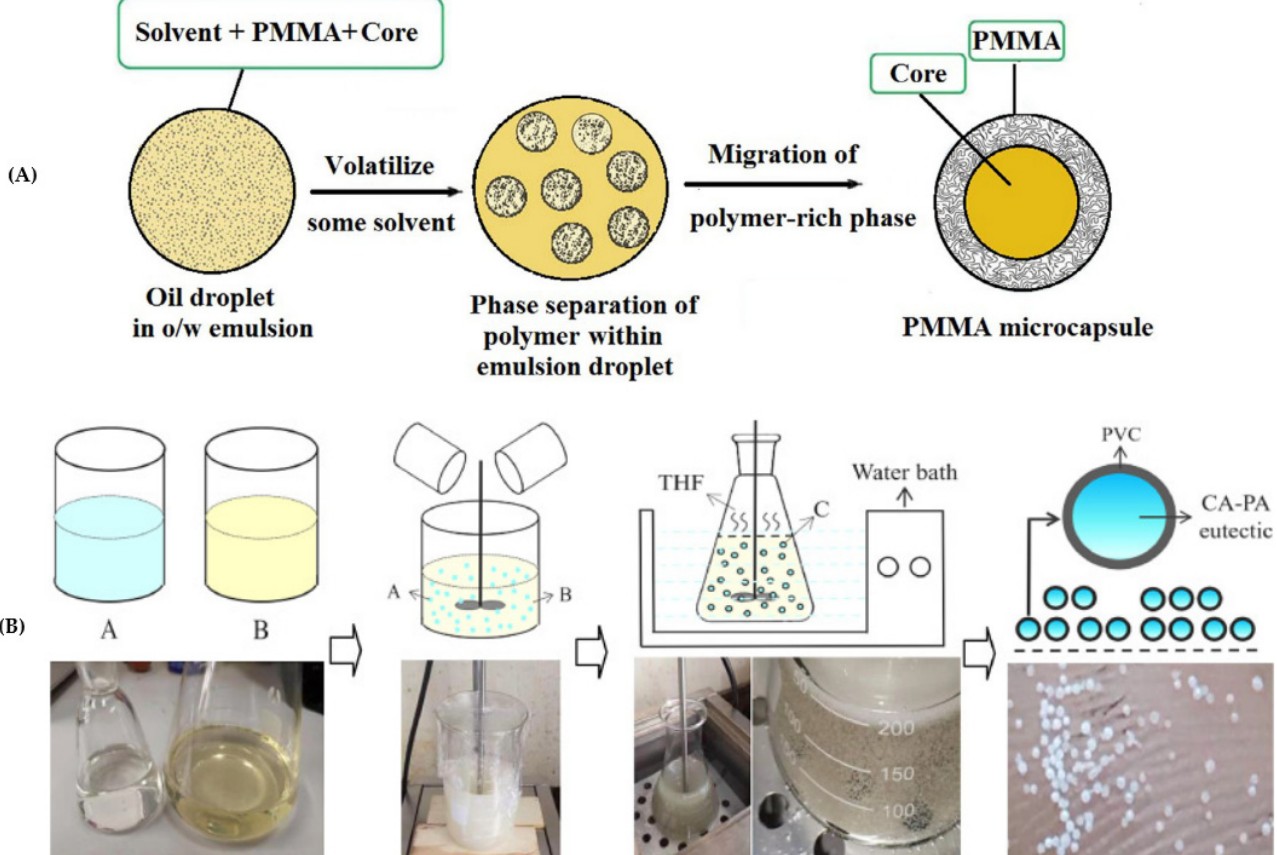

**Figure 24.** (**A**) scheme representation of mPCMs preparation by means of solvent evaporation and (**B**) experimental steps of microencapsulation of fatty acid eutectic (CA-PA) with PVC shell. Reused from Ahangaran et al. (2019) [104] and Xing et al. (2021) [65].

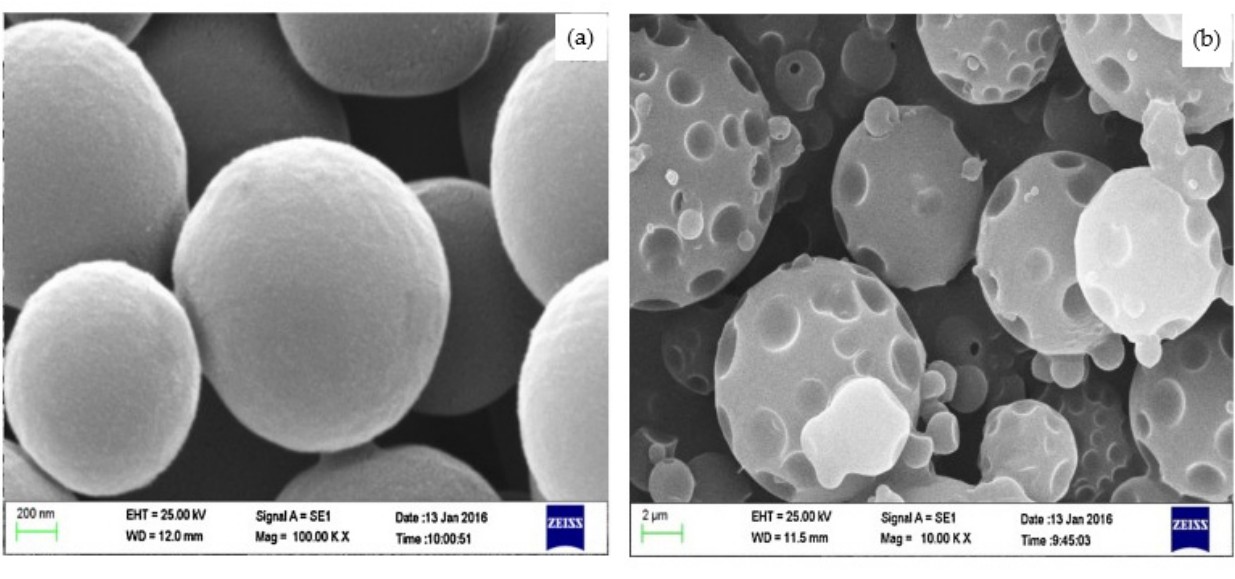

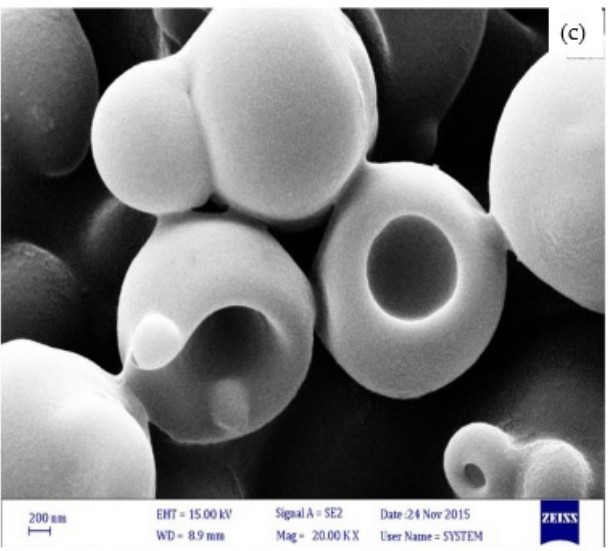

**Figure 25.** SEM micrographs of healing agent/PMMA microcapsules with: (**a**) core/shell, (**b**) acorn-shaped, and (**c**) golf ball-like morphologies. Reused from Ahangaran et al. (2017) [170].

Xing et al. (2021) [65] prepared a fatty acid eutectic ((CA-PA) mPCMs using polyvinyl chloride (PVC) as a shell material. The microcapsules were prepared by the solvent evaporation method and tetrahydrofuran (THF) was used as a solvent. The as-prepared mPCMs with a core-to-shell mass ratio of 2:1 had a latent heat of 92.1 J/g, corresponding to a PCM content of 57.7 wt.%. Additionally, the mPCMs showed an excellent thermal reliability following 500 thermal cycles. Microcapsules containing sodium phosphate dodecahydrate (inorganic PCM) were prepared using the solvent evaporation precipitation method [171]. Modified PMMA microcapsules containing disodium hydrogen phosphate heptahydrate ($Na_2HPO_4 \cdot 7H_2O$) were prepared by means of the suspension copolymerization solvent volatile method [172]. This method is not commonly used for the fabrication of mPCMs due to the nature of the porous shell formed and low PCM mass content. A summary of thermophysical properties of mPCMs synthesized using different microencapsulation methods is reported in Table 4.

**Table 4.** Summary of some prepared mPCMs with other methods.

| Method | Core | Shell | $T_{mp}$ (°C) | PS (μm or/nm) | $LH_m$ (J/g) | CC (wt.%) | Ref. |
|---|---|---|---|---|---|---|---|
| Complex coacervation | oxalic acid dihydrate | EC-ABS | 91.3 | 57.7–95.3 μm | 178.4 | 61.9 | [141] |
| Complex coacervation | sugarcane wax−$Al_2O_3$ composite | gelatine−gum Arabic | 70.7 | n/a | 59.7 | 87.6 | [173] |
| Complex coacervation | n-eicosane | SF-CHI | 38.0 | 23 μm | 149.5 | 61.8 | [51] |
| Spray drying | Rubitherm®RT27 | LDPE-EVA copolymer | 29.4 | n/a | 98.1 | 49.3 | [63] |
| Electro-Spraying | n-hexadecane | Polycaprolactone | 18.0 | 21.5 ± 3.1 μm | 113.2 ± 7.2 | 81.1 ± 5.2 | [174] |
| Sol–gel method | Paraffin | $SiO_2$ | 48–50 | n/a | 161.4 | 80.0 | [175] |
| Sol–gel method | lauric acid | $SiO_2$ | 44.2 | n/a | 186.6 | 78.6 | [176] |
| Sol–gel method | Paraffin | $GO/TiO_2$ | 51.53 | n/a | 160.8 | 81.9 | [58] |
| Self-assembly | Stearic acid | $CaCO_3$ | 56.6 | n/a | 161 | 91.2 | [177] |
| Self-assembly | n-eicosane | $CaCO_3$ | 37.2 | 740 nm–1.54 μm | 100–131.5 | 44.8–58.9 | [178] |
| Self-assembly | n-tetracosane | $CaCO_3$ | 51.0 | n/a | 134.0 | 53 | [179] |
| Self-assembly | n-eicosane | $Cu_2O$ | 38.70 | n/a | 165.3 | 61.6 | [164] |
| Self-assembly | palmitic acid | $CuCO_3$ | 66.9 | n/a | 89.3 | 43.9 | [180] |
| Microfluidic method | n-hexadecane | cellulose acetate | 18.0 | 88.3 μm | 176.0 | 66.0 | [169] |
| Microfluidic method | n-octadecane | $Fe_3O_4$-polyurea | 29.3 | 35–500 μm | 165.7 | 83.6 | [68] |
| Solvent evaporation | myristic acid | ethyl cellulose | 54.7 | n/a | 122.6 | 60.0 | [64] |
| Solvent evaporation | CA-PA eutectic | polyvinyl chloride | 17.1 | n/a | 92.1 | 57.7 | [65] |

Abbreviations: Ethyl cellulose (EC), acrylonitrile-butadiene-styrene (ABS), silk fibroin (SF), chitosan 802 (CHI), low-density polyethylene (LDPE), ethyl vinyl acetate (EVA), graphene oxide (GO), calcium 803 carbonate ($CaCO_3$), cuprous oxide ($Cu_2O$), Copper carbonate ($CuCO_3$), Iron oxide (Fe3O4), not available (n/a).

### 2.2.7. Microencapsulation of Inorganic PCM

The majority of studies involving the microencapsulation of PCMs have focused mainly on encapsulating organic-based PCM, with only a few currently focusing on the encapsulation of inorganic PCMs at the microscale. The reason for this is traced to the hydrophobic nature of organic PCMs [181]. Ideally, the two solvents constituting the core and shell parts of the microcapsules should not be miscible. This makes it possible to form the desired o/w emulsion prior to the polymerization process. Being able to microencapsulate PCM improves the handling, enhances compatibility of the PCM with the surrounding as the shell serves as a barrier, and enhances the heat transfer of the PCM resulting from a larger surface area compared to its volume. It is noteworthy to mention that inorganic PCMs are classified as salt hydrates, salts, and metals. A key advantage to that their usage as a thermal energy storage material is because they possess higher phase change enthalpy. The downside to their usage is that most inorganic PCMs exhibit features such as corrosion, subcooling, phase segregation, thermal instability, and phase segregation. Notwithstanding, some recent studies have reported the microencapsulation of inorganic PCMs through the solvent extraction–evaporation approach. In one study, sodium phosphate dodecahydrate (core) was microencapsulated with a shell material which was made of various organic solvents, cellulose acetate butyrate, and methylene diisocyanate as a cross-linker. The study achieved the microencapsulation process through solvent evaporation–precipitation technique. According to the authors, the final mPCMs' surface morphology was largely influenced by the nature of the solvent used (Salaun et al. [171]). The suspension copolymerization/solvent volatile technique was adopted in a study by Huang et al. [172] for the microencapsulation of inorganic disodium hydrogen phosphate heptahydrate using modified PMMA as the polymeric shell material with different organic solvents. The microencapsulated $Na_2HPO_4 \cdot 7H_2O$, having a mean diameter of about 6.8 μm, was verified as the core of the mPCM resulting from the dehydration of the $Na_2HPO_4 \cdot 12H_2O$ precursor.

When the microcapsules were subjected to heating within a temperature range of 30–84 °C, a mass loss of <10% was recorded, which is considered too high. The DSC results also revealed that the PMMA/ $Na_2HPO_4 \cdot 7H_2O$ had a melting temperature of approximately 51 °C and latent heat of 150 J/g. With the loss of about 10%, the inorganic PCM salt is highly thermally unstable, especially when tested for the long term. Hassabo et al. [182] investigated the microencapsulation of various metal PCM salts such as calcium nitrate tetrahydrate, $CaCl_2 \cdot 6H_2O$, $Na_2SO_4 \cdot 10H_2O$, disodium hydrogen phosphate dodecahydrate, ferric nitrate nonahydrate ($Fe(NO_3)_3 \cdot 9H_2O$), and magnesium (II) nitrate hexahydrate in tetraethoxysilane (silica) shells via polycondensation. The technique adopted was based on dis-solving silica in toluene and mixing the resultant dispersion in the hydrated salt, and through ultrasonic emulsification, a Pickering emulsion was formed and about 20% of poly(ethoxysiloxane) in toluene was added in a dropwise manner. According to the authors, only mPCMs made with $Na_2SO_4 \cdot 10H_2O$ and $Na_2HPO_4 \cdot 12H_2O$ as core materials displayed good phase change properties when incorporated in the polypropylene film polymer matrix. However, a thermal cycling test which would provide an indication of the thermal stability of the fabricated composite was not conducted in the study.

To further elucidate the microencapsulation of inorganic PCM, a particle fluidization process for the microencapsulation of two inorganic PCMs was considered for the microencapsulation of magnesium chloride hexahydrate ($MgCl_2 \cdot 6H_2O$) and bischofite polymer as the core and acrylic as the shell material [67]. The solvent used was chloroform and the preferred atomization flowrate was 2 kg/h at an atomization time of 2 min. In this process, crystal particles are suspended in an air flow system inside a fluidization column (Figure 26). A tabular illustration of the visual and microscopic observation of the pure inorganic PCM and as-produced microcapsules viewed with a microscope are presented in Table 5. The presence of more whitish-colored particles in the capsules is indicative of the degree of encapsulation achieved.

**Table 5.** Visual and optical microscopic images of the pure inorganic PCM and as-prepared microcapsules. Reprinted/adapted with permission from [67]. 2016, Svetlana Ushak, M Judith Cruz, Luisa F Cabeza and Mario Grágeda".

| Material | $MgCl_2 \cdot 6H_2O$ | Bischofite |
|---|---|---|
| PCM |  |  |
| PCM Microcapsule |  |  |
| Microscopic view X10 |  |  |

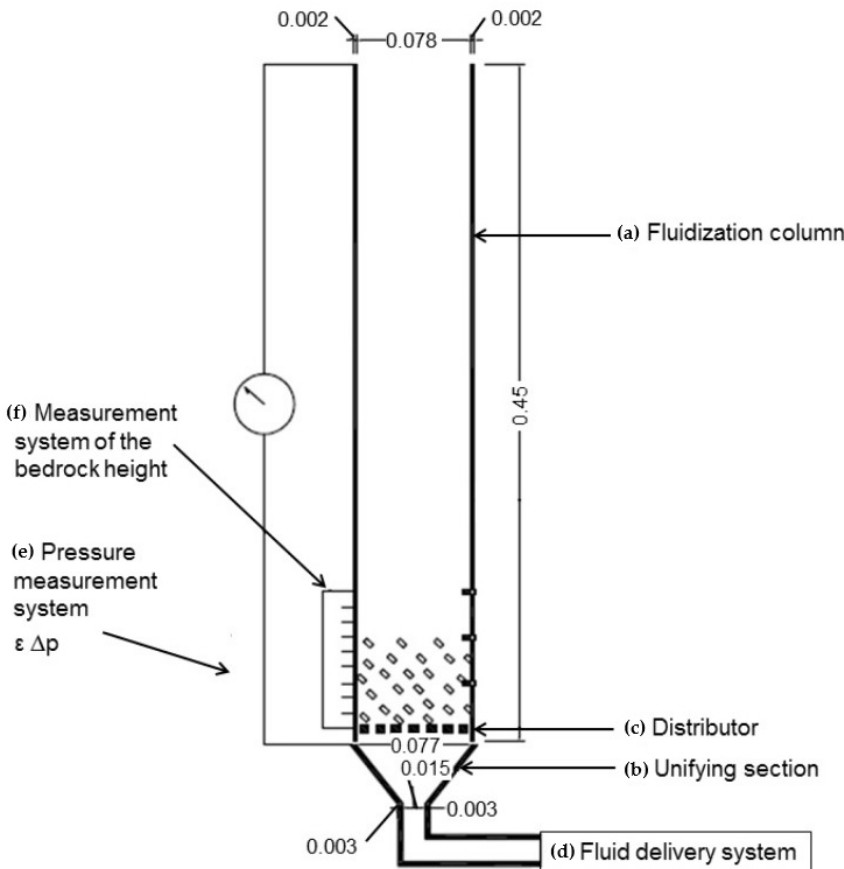

**Figure 26.** A schematic diagram of the fluidization column used in the microencapsulation of inorganic PCMs. Reprinted/adapted with permission from [67]. 2016, Svetlana Ushak, M Judith Cruz, Luisa F Cabeza and Mario Grágeda.

The thermal analysis result showed that both the $MgCl_2 \cdot 6H_2O$/acrylic and Bischofite/acrylic microcapsules had an average latent heat and melting temperatures of 106.8 J/g and 61 °C, and 99.8 J/g and 80.3 °C, respectively. The encapsulation efficiency reported was within the range of 87.02% and 92.22%, which is considered very high. A summary of the advantages and disadvantages of the common method used for PCMs' microencapsulation is presented in Table 6.

**Table 6.** Summary of advantages and disadvantages of common methods used for PCMs microencapsulation.

| Method | Advantages | Disadvantages |
|---|---|---|
| In situ polymerization | - capable of producing mPCMs with high storage capacity<br>- mild processing conditions<br>- achieve reasonable encapsulation efficiency<br>- good thermal/chemical stability<br>- spherical capsules with uniform coating<br>- less supercooling<br>- good mechanical resistance | - used toxic shell materials e.g., formaldehyde-based microcapsules<br>- needs to adopt a significant preparation route coupled with stringent safety precaution measures<br>- limited applications due to the some of the residues of formaldehyde resins in shells which can cause environmental and health problems |

**Table 6.** *Cont.*

| Method | Advantages | Disadvantages |
|---|---|---|
| Interfacial polymerization | - Monodisperse particle size distribution<br>- Easy to control<br>- less supercooling<br>- reasonable storage density | - fragile shell microcapsules<br>- high mass loss (PCM leaking through the shell)<br>- low durability |
| Suspension polymerization | - good energy storage density<br>- high mechanical resistance<br>- long-term durability<br>- environmentally friendly shell material used<br>- low-cost production<br>- easy to scale-up | - some monomers diffuse to aqueous phase and produce solid particles (empty of PCM)<br>- problem of supercooling; needs to add a proper nucleating agent |
| Emulsion/mini emulsion polymerization | - nano-microcapsules production<br>- uniform coating and morphological structure<br>- encapsulated bio-based PCM<br>- rapid encapsulation | - low encapsulation efficiency |
| Coacervation | - used green shell materials<br>- produced uniform particle size<br>- easy to control | - capsules agglomeration<br>- shell materials are host of bacteria growth<br>- limited to scale up |
| Sol–gel method | - good energy storage capacity<br>- high thermal conductivity<br>- can encapsulate inorganic PCMs | - wall permeability (pours shell structure)<br>- precursors used for producing mPCMs are relatively expensive and unattractive.<br>- PCM evaporation (mass loss) |
| Spray drying | - easy to scale up<br>- easy to control<br>- availability of equipment | - particles agglomeration<br>- wall permeability (PCM mass loss)<br>- not suitable for encapsulation of heat-sensitive materials |

## 3. Thermophysical Enhancement of mPCMs

Parameters such as energy storage, chemical and thermal stability, thermal conductivity, shell mechanical strength, and material resistance to fire are useful in examining the thermophysical nature of microcapsules [183–185]. However, common issues, including low energy density, high flammability, and poor thermal conductivity, of these fabricated mPCMs has limited their applicability. Consequently, enhancing the desired characteristics of mPCMs is a requirement to increasing their performance and widen their applications. In the following subsections, a brief outline on the most common approaches used for thermophysical enhancement of mPCMs is discussed.

### 3.1. Heat Transfer Enhancement

Thermal conductivity reflects the ability of heat transfer and is essential to the effective charge/discharge rates of PCMs. PCMs are poor conductors of heat with their thermal conductivity ranging between 0.1 and 0.4 W/(m·K). For instance, n-octadecane has a solid-state thermal conductivity of 0.35 W/(m·K) and a liquid-state conductivity of 0.149 W/(m·K). The low thermal conductivity of PCM does not limit its use in large-scale applications, e.g., buildings, but may become a crucial factor when applied directly, as mPCMs or as slurries in dynamic applications, such as in the cooling of electronics devices. Increasing the heat transfer for PCM charging/discharging can be achieved by confining the PCM in a tiny container known as a microcapsule. However, the encapsulated PCM with polymer shell is limited by its characteristic poor thermal conductivity. The pres-

ence of inorganic materials in the shell could produce PCM microcapsules with excellent properties. Wang et al. (2018) [186] synthesized a new type of microcapsule based on n-octadecane core and PMF/silicon carbide (SiC) hybrid shell for thermal energy storage and heat-transfer enhancement. The incorporation of nano-SiC (7% nano-SiC) suppressed the supercooling crystallization and improved the thermal transfer properties of the PMF/SiC mPCMs, which is reflected through the increase in thermal conductivity by 60.34%. Do et al. (2021) [187] improved the thermal conductivity of mPCMs through encapsulating the n-eicosane (PCM) with high thermal conductive hybrid shell materials, $Fe_3O_4@SiO_2@Cu$. The n-eicosane-$Fe_3O_4@SiO_2@Cu$ microcapsule has an excellent heat transfer ability owing to its high thermal conductivity, where the effective thermal conductivity of pure n-eicosane increases from 0.4716 to 1.3926 W/(m·K) when it is encapsulated with $Fe_3O_4@SiO_2@Cu$ shell. In particular, the thermal conductivity of the 50 nm mPCMs sample (1.3926 W/(m·K)) was higher than that of the 10 nm mPCMs sample (1.1469 W/(m·K)). This means that a larger amount of Cu metal NPs was loaded into the capsule, as observed in the TEM image, and that a larger metal NP size results in better thermal conductivity, as shown in Figure 27.

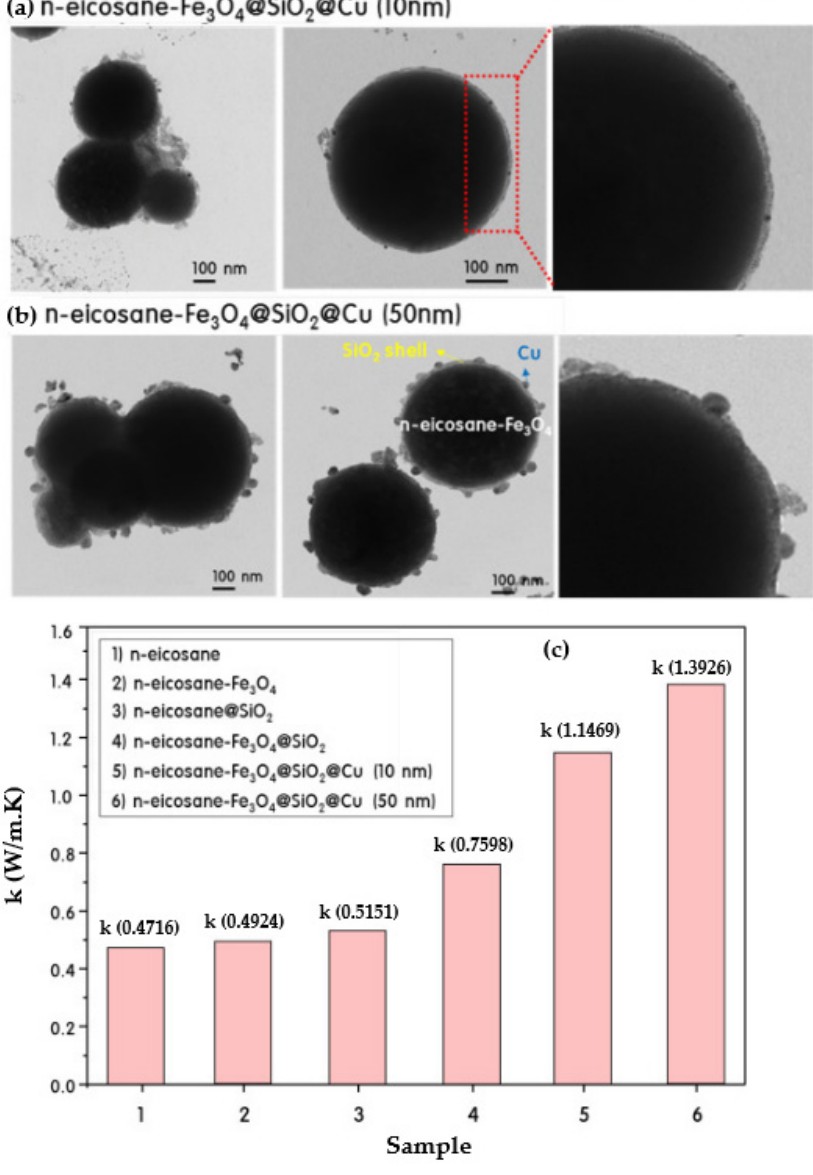

**Figure 27.** TEM images of the microencapsulated (**a**) n-eicosane-$Fe_3O_4@SiO_2@Cu$ (10 nm) and (**b**) n-eicosane-$Fe_3O_4@SiO_2@Cu$ (50 nm) samples and (**c**) thermal conductivity (k (W/m·K)) of the n-eicosane, and mPCMs samples. Reused from Do et al. (2021) [187].

Further investigations on producing mPCMs with inorganic–organic hybrid shells were recently reported in the literature. Li et al. (2019) [188] used a PMMA/TiO$_2$ hybrid shell to encapsulate n-octadecane through a facile emulsion method. The aggregation process was performed by an eco-friendly and efficient way under a UV radiation source for 10 min at 35 °C. Cheng et al. (2022) [189] prepared a novel Fe$_3$O$_4$/carbon nanotubes (CNTs)-modified PCM microcapsule (Fe–C-PCM) by the in situ polymerization method. The results showed that the CNTs in Fe–C-PCM acted as internal and external heat transfer channels to ameliorate the thermal conductivity of PMMA. The coated-polymer encapsulated PCMs with inorganic shells were also investigated. Similarly, Al-Shannaq et al. (2016) [190] developed a simple method of metal coating of mPCMs utilizing the following routes: (a) microcapsules surface functionalization using polydopamine (PDA), and (b) electroless plating of metallic silver. The as-prepared mPCMs were covered with Ag metal when they were pretreated with PDA as indicated by SEM in Figure 28a,b. Additionally, EDX surface-components analysis confirms the existence of Ag on the surface of the mPCMs (Figure 28c). The addition of metal coating on the bulk compacted mPCMs led to an increase in the thermal conductivity from 0.189 to 2.41 W/(m·K). The degree of Ag coating coverage on the mPCMs' surfaces with a rapid increase upon the formation of continuous thermal conductive pathways was responsible for the enhancement in thermal conductivity recorded. Subsequently, as shown in Figure 28d, after full surface coverage, there was no apparent increase in the thermal conductivity. In another study, Mikhaylov et al. (2019) [191] investigated the coating of PCMs with a zinc oxide polymeric shell. It was shown that hydrogen peroxide sol–gel processing holds promise in forming such uniform zinc peroxide coatings, which are then converted by chemical treatment with sodium sulfate to zinc oxide shell. The PCM microcapsules were completely covered with zinc oxide so the thermal conductivity and diffusivity of the PCM microcapsules were enhanced.

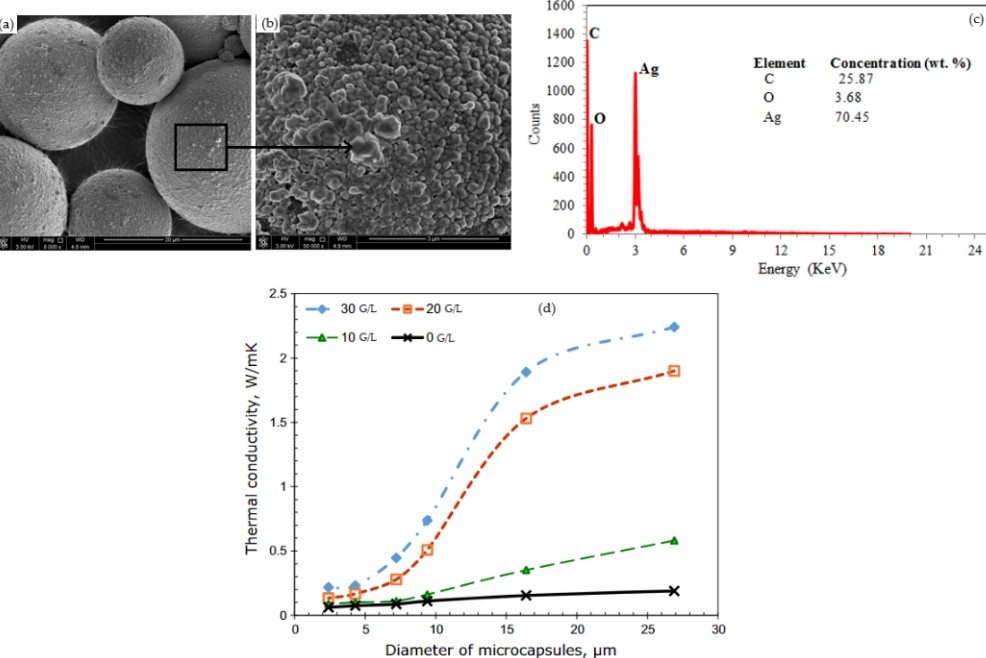

**Figure 28.** SEM images of metal-coated mPCMs with different magnifications (**a**,**b**), (**c**) EDX patterns of metal-coated mPCMs, and (**d**) measured apparent thermal conductivity versus mean diameter of the mPCMs at different silver nitrate concentration. Reused from Al-Shannaq et al. (2016) [190].

### 3.2. Supercooling Suppression

Supercooling is a common phenomenon in some thermal energy storage materials in which a material in its liquid state tends to solidify below its ideal freezing point temperature. For PCMs, supercooling limits their widespread applications. Salt hydrates

(an example of inorganic PCMs) are known to exhibit significant supercooling degree [192], while paraffins (also known as organic PCMs) do not experience serious subcooling [12]. Nevertheless, the microencapsulation of organic PCM tend to result in severe supercooling, which may be due to the absence of nuclei in such a small space. Hence, proper knowledge of the crystallization process is essential to advancing PCM research and technology. PCM crystallization undergoes several phases, such as an induction phase, crystal growth phase, and crystal regrowth phase. During the induction phase, nuclei are formed and grow to a sufficient size to be stable (nucleation centers are formed). Next, crystalline PCM diffuses toward the nucleus to be adsorbed on its surface, and hence grows the nucleus. The adsorbed material migrates along the surface and is incorporated into a crystal form. These small crystal seeds continue to grow until they become large enough to sustain a rapid rate of crystal growth. As the freezing process nears completion, the rate of crystallization tends to slow down. However, the redistribution of crystals continues to modify the shapes and particle size distribution, even after complete solidification of the material had occurred [46]. The time–temperature curves of $H_2O$ during freezing and supercooling processes is illustrated in Figure 29.

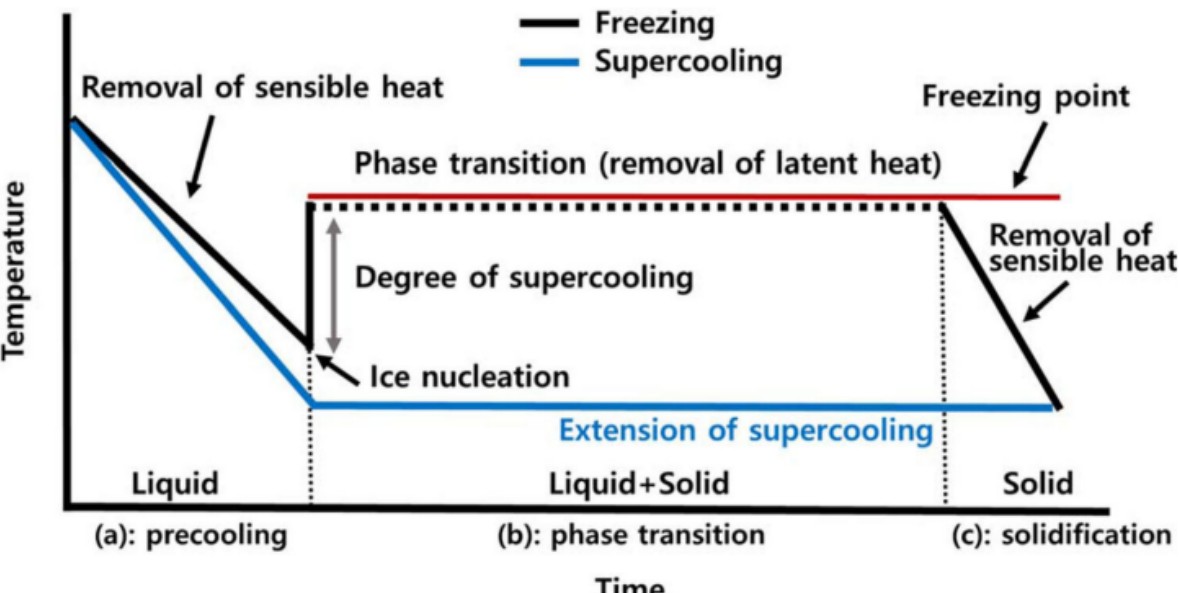

**Figure 29.** Typical time–temperature profiles of water during freezing and supercooling processes. Reused from Kang et al. (2020) [193].

Researchers have followed different strategies to eliminate or reduce the degree of supercooling of mPCMs. For example, Lei et al. (2022) [194] suppressed the mPCMs supercooling by loading nanoparticles into the core material. Three types of nanoparticles were used, including nano-BN, nanodiamond, and nano-Fe. The results demonstrated that nano-Fe with a concentration of 0.9% and particle size of 50 nm showed the best supercooling suppression effect, whereas the supercooling degree of mPCMs decreased by 41.5% in comparison with original the mPCMs sample (unloaded nanoparticles). Cao and Yang (2014) [195] investigated the effect of mPCMs' shell compositions on the degree of PCM supercooling. The results showed that without the inclusion of nucleating agents, a homogeneous nucleation is facilitated by shell-induced nucleation of the triclinic phase and the metastable rotator phase when the shell compositions and structure are optimized. Accordingly, the onset of freezing temperature shifted from 21.3 to 23.4 °C when the mass ratio of formaldehyde to melamine was 1.25 at a pH of 8.5. Another potential approach in eliminating mPCM supercooling is the addition of nucleating agents or metal additives to the PCM prior to encapsulation. Consequently, in their bid to eliminate mPCM supercooling, Al-Shannaq et al. (2015) [46] used Rubitherm® RT58 and 1-octadecanol as nucleating

agents. Furthermore, different types of nucleating agents, such as 1-tetradecanol, sodium chloride, 1-octadecnol, and paraffin were tested [132,195–197]. Adding the nucleating agent, which has a freezing point higher than that of the PCM, plays an important role in eliminating PCM supercooling. However, a drop in the latent heat of the microcapsules was observed by the addition of a large quantity of non-PCM material, which was unfavorable.

To synthesize mPCMs with low-supercooling abilities and high energy-storage densities, Wang et al. (2021) [198] designed novel microencapsulated composite phase change materials (CPCMs) via in situ polymerization. The CPCMs were composed of low-cross-linked polyoctadecyl methacrylate (C-PODMA), MF resin as the shell, and n-octadecane as PCM. The C-PODMA served two purposes, as a supporting matrix and as energy storage material. During the synthesis process, the crystal transformation and heterogeneous nucleation caused by C-PODMA can inhibit the supercooling degree of CS-MCPCMs to some extent, which may decrease from 8.3 °C to 3.7 °C and increase the crystallization temperature from 1.0 °C to 2.6 °C. Concurrently, the thermal decomposition temperature of CPCMs increased from 11.6 °C to 34.6 °C, thus enhancing the energy storage efficiency of CS-MCPCMs to 84.2%.

### 3.3. Mechanical Strength

In practical applications, especially in buildings, the candidate microcapsule shells must be robust enough to preclude rupture when they are being processed alongside other building materials. The mPCMs are also used in slurries to improve thermal performance of heat exchangers, and these microcapsules need to sustain the high shear caused by pumping the suspension. The mechanical strength of the mPCMs is determined by their chemical composition, structure, size, and shell thickness. In several studies, carbon nanotube has been used as part of microcapsules' shell materials to enhance the temperature response of PCMs as well as the capsules' shell mechanical strength [199–201]. A new type of mPCM with double-walled shells (MF resin/CNT-poly(4-styrenesulfonic acid) sodium) was developed through a facile layer-by-layer self-assembly technique using prefabricated MF resin-encapsulated n-octadecane as the template [202]. The self-assembly schematic representations of CNT-hybrid MPCMs containing various numbers of (PSS/A-CNT) bilayers are shown in Figure 30a. The SEM images confirmed the incorporation of CNTs within MF resin shell (Figure 30d–f). Additionally, the incorporation of 3.34 wt.% CNTs resulted in mPCMs with an average hardness and Young's modulus of 3.3 and 1.3 times greater than those of microcapsules without CNTs, respectively (Figure 30h). Furthermore, the incorporation CNTs with MF resin enhanced the temperature response, heat transfer, and thermal stability of mPCMs (Figure 30i).

Su et al. (2005) [203] successfully prepared double-shell-structured microcapsules containing PCMs with an average diameter of 5–10 μm using melamine–formaldehyde resin as the shell. The mechanical properties of single and double polymeric shell PCM microcapsules were tested under compression using a pressure sensor. Stress vs. strain curves showed that both single- and double-shell PCM microcapsules had a conventional plastic behavior when the applied pressure was beyond the yield strength of the shell. The reported results show a large variation in the mechanical strength of the tested single-shell PCM microcapsules with a yield point range from 30 to 40 kPa and a burst point range from 90 to 120 kPa. These variations indicate that the shell formed is not structurally uniform. However, the double polymeric shell mPCMs showed much more consistency in the yield point and burst point of the order of 55 and 134 kPa, respectively. The double shell mPCMs were structurally homogeneous with a better mechanical stability than that of single-shell mPCMs [204]. A methanol-modified melamine-formaldehyde (MMF) shell was used to enhance the mechanical properties of the MF shell [85]. The rigidity of the MMF shell was tested by placing the microcapsules between two pieces of glass and compressing them. A pressure sensor under the bottom glass measured and recorded the pressure intensity. The results show that the microcapsule's shell undergoes a plastic deformation when the applied pressure was beyond the yield point of the shell material. The MMF shells show

enhancement in the mechanical properties in comparison to the MF shells at the same core–shell mass ratio.

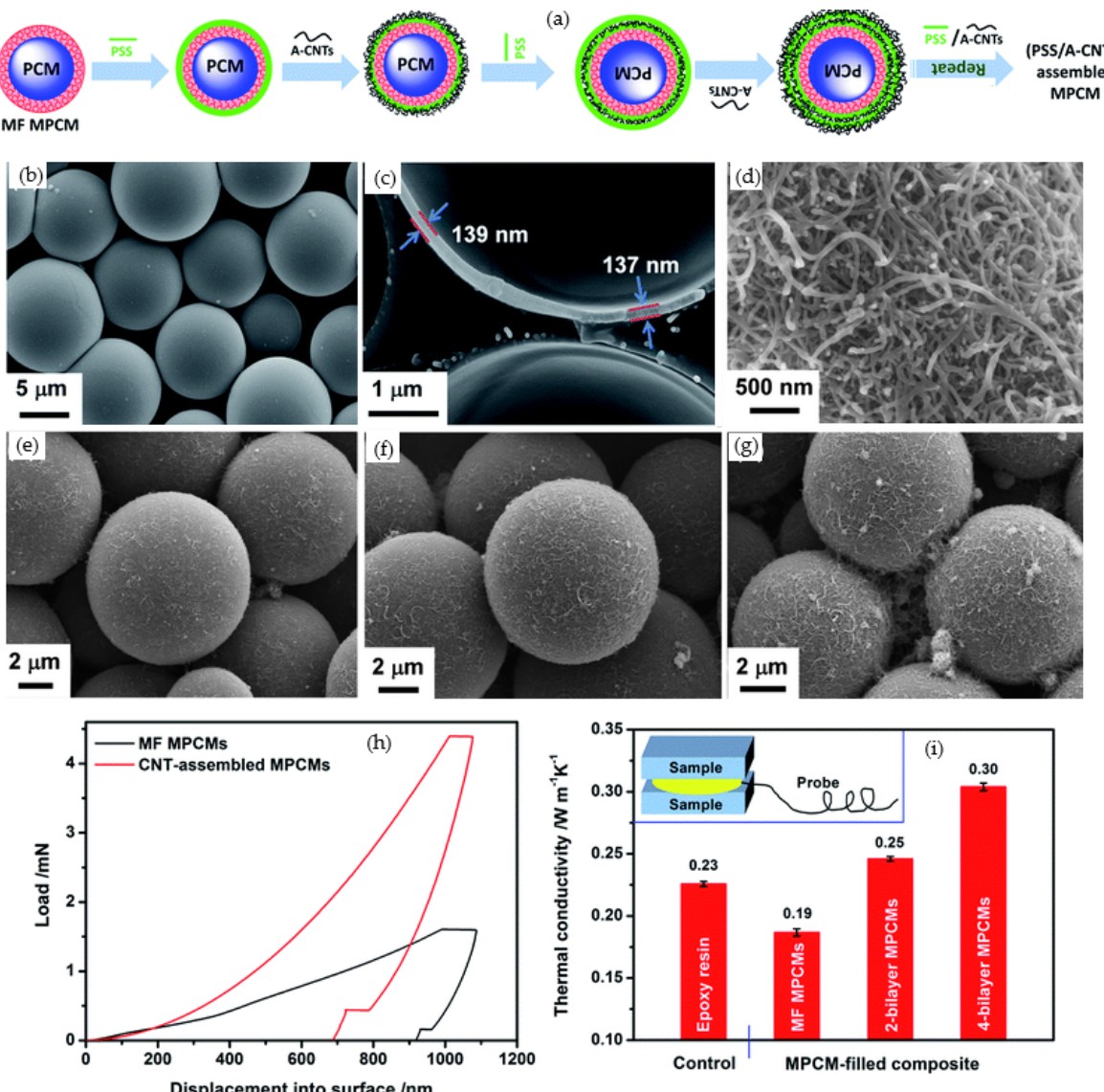

**Figure 30.** (**a**) The self-assembly process of CNT-hybrid MPCMs containing various numbers of (PSS/A-CNT) bilayers, SEM images, (**b**) MF mPCMs, (**c**) microcapsule cross section, (**d**) A-CNTs, (**e**) two bilayers CNTs mPCMs, (**f**) three bilayers CNTs mPCMs, (**g**) four bilayers CNTs mPCMs, (**h**) load-displacement curves of MF MPCMs and CNT-assembled MPCMs with four (PSS/A-CNT) bilayers, and (**i**) thermal conductivity of mPCMs with and without CNTs. Reused from Huang et al. (2017) [202].

Controlling the size of the PCM microcapsules is considered a realistic option for improving shell mechanical strength. The force required to rupture a single microcapsule of commercial product (Microtek MPCM-18D and Micronal® DS 5008 X (BASF)) with different sizes was investigated [44,168]. The nanocompression test of a single microcapsule was performed using MTS XP Nano-indenter with 10μm radius 60° conical diamond tip and 40-X optical microscope lens. The results showed that the MPCM-18D with size diameters between 18.75 and 28.3 μm required a force of 0.67 mN and 0.29 mN to rupture, respectively; thus, smaller microcapsules required greater force to rupture. Unsaturated monomers such as methyl methacrylate can be tuned with cross-linking agents to produce polymers with high mechanical strength. Cross-linked PMMA microcapsules containing n-octadecane

were successfully prepared using the suspensionlike polymerization method. Four cross-linking agents with different numbers of cross-linking functional moieties were selected for comparison. The results show that the PCMs' microcapsules prepared using a high number of cross-linking functional moieties exhibited a greater shell mechanical strength [205,206].

## 4. Conclusions and Future Trends

PCM microcapsules widen the application areas of the PCMs. Microencapsulation has been shown to provide effective containment of PCM through increased heat transfer area, reduce PCM reactivity to the external environment, and most importantly, prevent PCM exudation. Currently, there are numerous methods available for mPCM preparation, each of which has its advantages as well as limitations. This review discussed, up to date, the different manufacturing approaches followed in preparing mPCMs. It also discussed the potential approaches used to enhance the thermophysical properties of the mPCMs, including heat transfer enhancement, supercooling suppression, and improving their shell mechanical strength. The thermophysical properties of mPCMs are heavily dependent on the raw materials and synthesis processes during microencapsulation. Shell materials of mPCMs play an important role in providing structural integrity, thermal stability, and PCM containment. The choice of shell materials depends on the PCM type, encapsulation process, and the end use. Interfacial, in situ, and suspension polymerizations are the common techniques for microencapsulation of PCM with polymer shell. The polymeric microcapsules showed excellent thermal and chemical stability, PCM reliability, sealing properties, ease of processing, and structural robustness. However, the applicability of mPCMs is limited by their flammability and poor thermal conductivity. Inorganic shells possess some desirable properties of being thermally conductive, having higher rigidity than the synthetic polymer, and being fire resistant. The current encapsulated PCM with an inorganic shell exhibits low encapsulation efficiency, with PCM leaking through the shell due to the nature of the porous network formed by the metal oxide. Currently, research on hybrid shells (organic and inorganic) is in its early stages, mainly focusing on trying various methods and materials to prepare mPCMs with good mechanical behavior, but with minimum success in obtaining good confinement and long-life mPCMs. Further synthesis of mPCMs with hybrid shells consists of many steps, which would hinder the production of mPCMs on a large scale. In conclusion, the successful application of a hybrid shell into the PCM microcapsules may solve the problems associated with synthetic polymer microcapsules and open the window for new applications.

As reported earlier, thermally driven polymerization can successfully encapsulate PCMs. However, polymerization time may take up to more than 4 h, which limits industrial production rates. In contrast, photo-induced polymerization with UV radiation offers a better alternative to thermal treatment by reducing the polymerization time and energy consumption. Another benefit of using UV radiation is that the microencapsulation process can be conducted at relatively low temperatures and thus can be used to encapsulate heat-sensitive PCMs. For example, in our previous publications [38,105,106], we successfully synthesized mPCMs using photo-induced polymerization through coiled-tube UV reactor. The use of a coiled-tube UV reactor with a sufficient amount of photoinitiation enables the encapsulation of heat-sensitive PCMs to be conducted at ambient conditions in a very short time. Photo-induced polymerization for microencapsulation of PCMs is rarely reported in the literature; therefore, further research and development is required to broaden the scope of PCM encapsulation, which could lead to a rapid, continuous, and potentially large-scale industrial production of cost-effective PCM microcapsules with low energy consumption. Focusing on the large-scale production of mPCM, information on process quality parameters, such as encapsulation efficiency, yield, and monomer conversion, is scarce, and as such, is recommended for future studies.

**Funding:** This research received no external funding.

**Conflicts of Interest:** The authors declare no conflict of interest.

## Nomenclature

| | | | | | |
|---|---|---|---|---|---|
| PCMs | Phase change materials | PDVB | Poly(divinylbenzene) | GEL | Gelatine |
| mPCMs | Phase change materials microcapsules | LDPE | Low-density polyethylene | MC | Methylcellulose |
| TES | Thermal energy storage | EVA | Ethyl vinyl acetate | XG | Xanthan Gum |
| UV | Ultraviolet | PVC | Poly(vinyl chloride) | PAM | Poly(acrylamide) |
| MMA | Methyl methacrylate | HEMA | Hydroxyethyl methacrylate | PEI | Poly(ethylenimine) |
| BA | Butyl acrylate | PDMS | Poly(dimethyl siloxane) | SDS | Sodium dodecyl sulfate |
| MF | Melamine-formaldehyde | ABS | Acrylonitrile-butadiene-styrene | LED | light emitting diode |
| UF | Urea-formaldehyde | CNC | Cellulose nanocrystal | OA | Oxalic acid |
| UMF | Urea-melamine formaldehyde | SF | Silk fibroin | TNBT | Tetra-n-butyl titanate |
| PEG | Polyethylene glycol | CHI | Chitosan | TEOS | Tetraethyl orthosilicate |
| PU | Polyurethane | EC | Ethyl cellulose | TEM | Transmission electron microscopy |
| TDI | Toluene-2, 4-diisocyanate | GO | Graphene oxide | EDX | Energy-dispersive detector |
| DETA | Diethylenetriamine | SEM | Scanning electron microscope | DSC | Differential scanning calorimetry |
| PBMA | Poly(butyl methacrylate) | TGA | Thermogravimetric Analysis | PA | Palmitic acid |
| PS | Poly(styrene) | SMA | Styrene Maleic anhydride | PW | Paraffin wax |
| HD | Heptadecane | TSCD | Trisodium citrate dihydrate | CTAB | Cetyltrimethylammonium bromide |
| HD | n-hexadecane | IPDI | Isophorone diisocyanate | DBTDL | Dibutyltin dilaurate |
| TEPA | Tetraethylenepentamine | PVA | Poly (vinyl alcohol) | CA | Caprylic Acid |
| THF | Tetrahydrofuran | SiC | Silicon carbide | CNTs | Carbon nanotubes |
| PDA | polydopamine | CPCMs | composite phase change materials | | |

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
