# Peer review of "Methods for the Synthesis of Phase Change Material Microcapsules with Enhanced Thermophysical Properties—A State-of-the-Art Review"

_2673-8023, doi:10.3390/micro2030028_

Round 1

Reviewer 1 Report

The paper is not complete, even if it could potentially be a good work. It needs to be improved with following major revisions:

1) The objective, methodology, and results should be better described, discussed and justified.

2) The current literature review is not sufficient. For example, please mention (give) following more current studies related to thermal energy storage by PCMs in the sections of Introduction, and References List for completeness of your study and the references:

- Development of a model compatible with solar assisted cylindrical energy storage tank and variation of stored energy with time for different phase change materials, Energy Conversion and Management, 37(12), 1775-1785 (1996).

 - Geometric design of solar-aided latent heat store depending on various parameters and phase change materials, Solar Energy, 62(1), 19-28 (1998).

 - Thermal performance of a solar-aided latent heat store used for space heating by heat pump, Solar Energy, 69(1), 15-25 (2000).

 3) The results should be expanded significantly and quantitatively.

4) I strongly suggest that authors shall carry out more studies to compare the results from this paper to that from other similar studies.

Author Response

Dear reviewer, 

We would like to thank you for taking the time to review our manuscript entitled “Methods for the synthesis of phase change materials micro-capsules with enhanced thermo-physical properties-A state-of-art review”, We appreciate your time taken to review the manuscript. Please find attached revised manuscript. Point-by-point response to reviewer is included (page 45). We have highlighted our response in red font.

Sincerely,

The Authors

Reviewer 2 Report

Methods of synthesis phase change materials microcapsules (mPCMs) with their enhanced thermo-physical properties were well reviewed. However, though the properties of microcapsules of inorganic PCMs were mentioned, there is no review on microencapsulation of inorganic PCMs. It is suggested that some papers on microencapsulation methods and their influence on the properties of microcapsules of inorganic PCMs should be added.

Author Response

Dear reviewer,

We would like to thank you for taking the time to review our manuscript entitled “Methods for the synthesis of phase change materials micro-capsules with enhanced thermo-physical properties-A state-of-art review”, We appreciate your time taken to review the manuscript. Please find revised manuscript. Point-by-point response is included (page 45). We have highlighted our response in red font. 

Sincerely,

The Authors

Reviewer 3 Report

This paper presents a review study on methods of synthesis phase change materials microcapsules together with their thermo-physical properties. Authors focused on providing critical guidance for selecting the proper synthesis method and materials for PCMs microencapsulation suitable for final product specifications.

Presented article may be interesting to the readership of this Journal. What is more, in my opinion, the research is complete and may be published after minor revision some few minor errors, which I wrote about below ("Minor errors" part).

In general
1) Introduction of the "nomenclature" part is recommended because there are some abbreviations and markings that should be defined in my opinion at the beginning of the paper.
2) Authors should think about a tabular summary of all methods, taking into account the advantages, disadvantages and application ranges of each. This would make searching for a specific method much easier.

Minor errors:
3) line 80: the unit should be "μm" instead of "um".
4) lines 201, 351, 370, 379, 843: the unit should be "°C" instead of "oC".
5) Quality of some figures (i.e. Fig.15, or Fig.20) are below the average. Please try to replace them with sharper ones.
6) lines 576, 665, 708, 816-7, 835-6, 854: the unit should be "W/(m·K)" instead of "W·m−1·K−1" or "W/m.K".
7) lines 585, 600, 607, 631, 635, 637, 639, 642, 644, 646, 648-9, 663, 669 673-680, 689, 697, 793, 831: in notations "SiO2", "TiO2" and other of this type, the number 2 should be written as a subscript.

Author Response

Dear reviewer,

We would like to thank you for taking the time to review our manuscript entitled “Methods for the synthesis of phase change materials micro-capsules with enhanced thermo-physical properties-A state-of-art review”, We appreciate your time taken to review the manuscript. Please find attached revised manuscript. Point-by-point response are included (page 45). We have highlighted our response in red font. 

Sincerely,

The Authors

Round 2

Reviewer 1 Report

The paper is acceptable for publication in its present form. The authors gave detailed answers to all of the previous comments and revised the manuscript accordingly. I think the paper has been significantly improved. Therefore my recommendation is that this paper can be published.